# Single cell transcriptomics identifies a unique adipose lineage cell population that regulates bone marrow environment

Leilei Zhong[1], Lutian Yao[1,2], Robert J Tower[1], Yulong Wei[1,3], Zhen Miao[4], Jihwan Park[5], Rojesh Shrestha[5], Luqiang Wang[1,6], Wei Yu[1,3], Nicholas Holdreith[7,8], Xiaobin Huang[8], Yejia Zhang[1,9,10], Wei Tong[7,8], Yanqing Gong[11], Jaimo Ahn[1], Katalin Susztak[5], Nathanial Dyment[1], Mingyao Li[4], Fanxin Long[12], Chider Chen[13], Patrick Seale[14], Ling Qin[1]*

[1]Department of Orthopaedic Surgery, Perelman School of Medicine, University of Pennsylvania, Philadelphia, United States; [2]Department of Orthopaedics, The First Hospital of China Medical University, Shenyang, China; [3]Department of Orthopaedics, Union Hospital, Tongji Medical College, Huazhong University of Science and Technology, Wuhan, China; [4]Department of Biostatistics, Epidemiology and Informatics, University of Pennsylvania Perelman School of Medicine, Philadelphia, United States; [5]Renal Electrolyte and Hypertension Division, Department of Medicine and Genetics, University of Pennsylvania, Philadelphia, United States; [6]Department of Orthopaedics, Shandong University Qilu Hospital, Shandong University, Jinan, China; [7]Division of Hematology, Children's Hospital of Philadelphia, Philadelphia, United States; [8]Department of Pediatrics, Perelman School of Medicine at the University of Pennsylvania, Philadelphia, United States; [9]Department of Physical Medicine and Rehabilitation, Perelman School of Medicine, University of Pennsylvania, Philadelphia, United States; [10]Translational Musculoskeletal Research Center (TMRC), Corporal Michael J. Crescenz Veterans Affairs Medical Center, Philadelphia, United States; [11]Division of Transnational Medicine and Human Genetics, Perelman School of Medicine, University of Pennsylvania, Philadelphia, United States; [12]Translational Research Program in Pediatric Orthopaedics, The Children's Hospital of Philadelphia, Philadelphia, United States; [13]Department of Oral and Maxillofacial Surgery/Pharmacology, University of Pennsylvania, School of Dental Medicine, Philadelphia, United States; [14]Department of Cell and Developmental Biology, Perelman School of Medicine, University of Pennsylvania, Philadelphia, United States

*For correspondence:
qinling@pennmedicine.upenn.edu

Competing interests: The authors declare that no competing interests exist.

**Abstract** Bone marrow mesenchymal lineage cells are a heterogeneous cell population involved in bone homeostasis and diseases such as osteoporosis. While it is long postulated that they originate from mesenchymal stem cells, the true identity of progenitors and their in vivo bifurcated differentiation routes into osteoblasts and adipocytes remain poorly understood. Here, by employing large scale single cell transcriptome analysis, we computationally defined mesenchymal progenitors at different stages and delineated their bi-lineage differentiation paths in young, adult and aging mice. One identified subpopulation is a unique cell type that expresses adipocyte markers but contains no lipid droplets. As non-proliferative precursors for adipocytes, they exist abundantly as pericytes and stromal cells that form a ubiquitous 3D network inside the marrow cavity. Functionally they play critical roles in maintaining marrow vasculature and suppressing bone

formation. Therefore, we name them marrow adipogenic lineage precursors (MALPs) and conclude that they are a newly identified component of marrow adipose tissue.

## Introduction

Osteoporosis is a silently progressive disease characterized by excessive bone loss and structural deterioration until it clinically presents as bone fragility and fracture. While mostly afflicting post-menopausal women and the elderly (*National Osteoporosis Foundation, 2002*), it also manifests in adolescents and adults as a result of endocrine and metabolic abnormalities or medications (*Mirza and Canalis, 2015*). Osteoporosis is largely or partially caused by diminished bone forming activity and often accompanied by increases in marrow adiposity. Therefore, understanding the nature of bone marrow mesenchymal stem cells (MSCs) and their differentiation routes into osteoblasts and adipocytes has always been the center of bone research. However, in contrast to the wealth of knowledge regarding hematopoiesis from hematopoietic stem cells (HSCs) (*Orkin and Zon, 2008*), our in vivo knowledge of MSCs and their descendants are largely incomplete, which has greatly limited advances in treating clinical disorders of bone loss.

It is well-accepted that bone marrow mesenchymal progenitors are heterogeneous, including MSCs and their descendants at various differentiation stages before they reach the terminal states as osteoblasts, osteocytes, and adipocytes. Our current knowledge of those progenitors is mostly dependent on cell expansion in culture for CFU-F, multi-lineage differentiation, and transplantation assays. However, in vitro culture does not necessarily replicate in vivo cell behavior after depleting their environmental cues. Another common approach is lineage tracing that uses a specific promoter-driven *Cre* or *CreER* system to label a portion of mesenchymal lineage cells. However, it generally does not provide information about the specific stage(s) of mesenchymal progenitors that cells start to be labeled. The recently available large-scale single cell RNA-sequencing (scRNA-seq), which is capable of identifying and interrogating rare cell populations and deducting the course of differentiation (*Wu et al., 2017*), finally provides an unbiased tool to investigate bone marrow mesenchymal cells in vivo.

Several recent reports applied this technique on mouse bone marrow mesenchymal cells. Based on previous studies that leptin receptor (Lepr) marks adult bone marrow MSCs (*Zhou et al., 2014*) and Lepr$^+$ cells serve as niche for hematopoietic progenitors (*Comazzetto et al., 2019*), one study used *Lepr-Cre* to label mesenchymal stromal cells and *Col1a-Cre* to label osteoblasts for analyzing HSC niches (*Tikhonova et al., 2019*). Another one utilized *Cxcl12-CreER* to label bone marrow stromal cells (*Matsushita et al., 2020*). The others depleted hematopoietic cells from bone marrow and analyze the remaining bone marrow cells (*Baccin et al., 2020*; *Baryawno et al., 2019*; *Wolock et al., 2019*). Interestingly, all those studies identified a large cell cluster expressing many adipocyte markers and in some studies this cluster was annotated as MSC. For our study, we used a different approach by taking advantage of a *Col2-Cre Rosa26 <lsl-tdTomato>* (Col2:Td) mouse model that we and others previously reported to label bone marrow mesenchymal lineage cells (*Ono et al., 2014*; *Chandra et al., 2017*). Since in this model Td labels every osteocyte and every bone marrow adipocyte in vivo and all CFU-F-forming cells, we reason that all bone marrow mesenchymal lineage cells, including all mesenchymal progenitors, are included within the Td$^+$ population. By applying large scale scRNA-seq on this population derived from various age groups, we identified an in vivo the most primitive progenitor population different from previous reports and delineated the in vivo evolvement of early mesenchymal progenitors into mature osteogenic and adipogenic cells through hierarchical differentiation paths. Among newly identified mesenchymal subpopulations, a new type of adipose lineage cells is subsequently validated and investigated for their critical actions in regulating bone marrow vasculature and bone homeostasis.

## Results

### Single cell transcriptomic profiling of bone marrow mesenchymal lineage cells

Consistent with our previous report (*Chandra et al., 2017*), Td labeled the following cells in the long bone of 1-month-old Col2:Td mice: all chondrocytes (1052 out of 1052 chondrocytes counted, n = 4

mice, 100%) in the growth plate, osteoblasts and all osteocytes (3600 out of 3600 osteocytes counted, n = 4 mice, 100%) in trabecular and cortical bone, all CD45$^-$ stromal cells (2050 out of 2050 cells with stromal morphology counted, n = 4 mice, 100%), all Perilipin$^+$ adipocytes (5 out of 5 adipocytes counted, n = 4 mice, 100%), and many pericytes throughout the bone marrow from metaphysis to diaphysis (*Figure 1A*, *Figure 1—figure supplement 1A*). We previously developed an enzymatic digestion method to release bone marrow cells trapped within the trabecular bone (endosteal bone marrow) and demonstrated that they contain a higher frequency of mesenchymal progenitors than central bone marrow flushed from the diaphysis of long bones (*Siclari et al., 2013*). This enzymatic digestion method collects osteoblasts as well and thus, is better than conventional flushing method by providing a definite cluster of mature cells for single cell analysis. Due to its small quantity, Td$^+$ peak among endosteal bone marrow cells was sometimes distinguishable (1.2 ± 0.1%) but often not. Nevertheless, fractioning all cells into the top 1%, 1–2%, and >2% cells based on the Td signal (*Figure 1—figure supplement 1B*) revealed that the top 1% group contains all the CFU-F forming cells (*Figure 1—figure supplement 1C*) and almost all CFU-F colonies (98.7%) are Td$^+$ (*Figure 1—figure supplement 1D*). Since osteoblasts are also Td$^+$, these data suggested that the top 1% cells include all bone marrow mesenchymal progenitors as well as osteoblasts.

Using 10x Genomics approach, we sequenced the top 1% Td$^+$ endosteal bone marrow cells from 1- to 1.5-mo-old Col2:Td mice. We profiled 13,759 cells with a median of 2686 genes/cell and a median of 12215 UMIs/cell (*Figure 1—figure supplement 2A*). Unsupervised clustering of the gene expression profiles using Seurat identified 22 groups, including 9 groups of mesenchymal lineage cells and 11 groups of hematopoietic cells, 1 group of endothelial cells, and 1 group of mural cells (*Figure 1—figure supplement 2B-D*). For reasons unclear at present, we observed a large number of non-mesenchymal lineage cells, which is confirmed by detectable *Td* and *Col2a1* expression in those cells (*Figure 1—figure supplement 3*). tSNE (*Figure 1B*) and UMAP (*Figure 1—figure supplement 4A*) analyses of 7585 mesenchymal lineage cells yielded the same cell cluster pattern containing 9 subpopulations. Cells from two batches of experiments were present at a roughly similar proportion in each subpopulation (*Figure 1—figure supplement 4B, C*). Examination of lineage-specific markers identified clusters with gene signatures of osteoblast (cluster 4), osteocyte (cluster 5), adipocyte (cluster 7), and chondrocyte (cluster 8 and 9, *Figure 1C*). Chondrocyte clusters, while unexpected, likely represent growth plate chondrocytes released from bone fragments during enzymatic digestion after we cut off epiphyses at the growth plate site. Since chondrocytes originate from the resting zone progenitors but not bone marrow MSCs (*Mizuhashi et al., 2018*), they were excluded from subsequent analysis.

Interestingly, pseudotemporal cell trajectory analysis using Monocle (*Trapnell et al., 2014*) placed cells in cluster 1 at one end of pseudotime trajectory and osteocytes (cluster 5) and adipocytes (cluster 7), two terminally differentiated cell types, at the opposite, divergent ends (*Figure 1D*). Slingshot (*Street et al., 2018*), another method for inferring cell lineages and pseudotimes, generated a similar trajectory pattern (*Figure 1E*). These data suggested that cluster 1 cells are the ancestor of other mesenchymal cells in the dataset and that they bi-differentiate into osteogenic and adipogenic lineage cells. Indeed, cluster 1 expressed several common stem cell markers, such as stem cell antigen 1 (Sca1), Cd34, and Thy1 (*Figure 1C*). Hence, we named cluster 1 as early mesenchymal progenitors (EMPs), the most primitive cells in the sequencing dataset. Since these data are derived from *Col2-Cre* labeled cells, we cannot exclude the possibility that the bone marrow contains even more primitive Td$^-$ cells that give rise to cluster 1 cells. That said, the labelling of essentially all CFU-F forming cells in Col2:Td bone marrow by Td, makes that possibility unlikely. Compared to cluster 1, clusters 2, 3, and 4 (osteoblast) expressed gradually increased levels of osteogenic genes, such as *Sp7*, *Runx2, Col1a1, Ibsp*, and *Bglap2*. Since clusters 2 and 3 are sequentially located after cluster 1 and before the branch point in pseudotime trajectory, we named cluster 2 as intermediate mesenchymal progenitors (IMPs) and cluster 3 as late mesenchymal progenitors (LMPs). Cells in cluster 6 were distributed around the branch point, indicating that they are lineage committed progenitors (LCPs).

Computational cell cycle analysis was performed to understand the proliferative status of each cluster (*Figure 1F*). As a positive control, chondrocyte cluster 9, corresponding to proliferative and prehypertrophic growth plate chondrocytes, had the highest proliferative status. It appeared that within bone marrow mesenchymal lineage cells, EMPs are less proliferative than IMPs; LMPs, LCPs, and osteoblasts are the most proliferative; adipocytes and osteocytes are non-proliferative. These

results further support our cluster annotation. Hierarchy analysis showed distinct gene expression signatures in each cluster (*Figure 1G*).

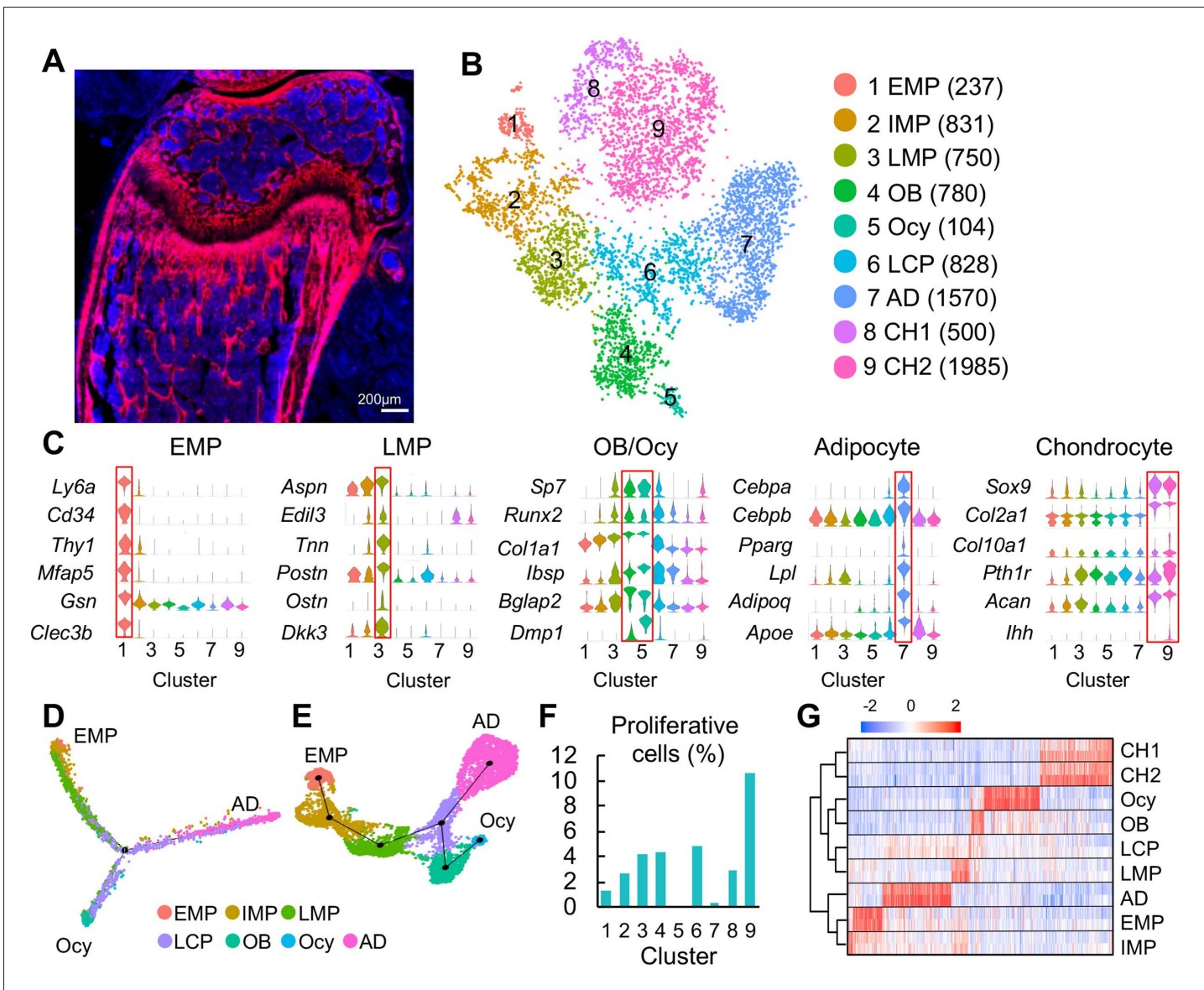

**Figure 1.** Clustering of bone marrow mesenchymal lineage cells by single cell transcriptomics reveals the in vivo identity of novel mesenchymal subpopulations. (**A**) Fluorescent image of distal femur of 1-month-old Col2:Td mice. (**B**) The tSNE plot of 7585 Td+ mesenchymal lineage cells isolated from endosteal bone marrow of 1–1.5 month-old Col2:Td mice (n = 5 mice). Cell numbers are listed in parenthesis next to cluster names. OB: osteoblast; Ocy: osteocyte; LCP: lineage committed progenitor; AD: adipocyte; CH: chondrocyte. (**C**) Violin plots of marker gene expression for EMPs, LMPs, osteoblast/osteocyte, adipocyte, and chondrocyte clusters. (**D**) Monocle trajectory plot of bone marrow mesenchymal lineage cells. (**E**) Slingshot trajectory plot of bone marrow mesenchymal lineage cells. In D and E, cells are labeled according to their Seurat clusters. (**F**) The percentage of proliferative cells (S/G2/M phase) among each cluster was quantified. (**G**) Hierarchy clustering and heatmap of mesenchymal lineage clusters. Color bar on the top indicates the gene expression level. Each cluster contains two batches (top and bottom) of samples.

The online version of this article includes the following figure supplement(s) for figure 1:

**Figure supplement 1.** Bone marrow Td+ cells from Col2:Td mice contain the entire set of mesenchymal lineage cells.

**Figure supplement 2.** Large scale scRNA-seq analysis of top 1% Td+ cells from endosteal bone marrow of 1–1.5 month-old Col2:Td mice.

**Figure supplement 3.** The expression patterns of *Tomato* and *Col2a1* in tSNE plots.

**Figure supplement 4.** No batch effect was detected in our analysis.

**Figure supplement 5.** The expression patterns of previously reported mesenchymal progenitor markers.

During the past decade, many proteins have been proposed as mesenchymal progenitor markers based on lineage tracing and in vitro culture data, such as Lepr (*Zhou et al., 2014*), myxovirus resistance-1 (Mx1) (*Park et al., 2012*), glioma-associated oncogene homolog 1 (Gli1) (*Shi et al., 2017*), paired related homeobox 1 (Prxx1) (*Logan et al., 2002*), gremlin (Grem1) (*Worthley et al., 2015*), platelet derived growth factor receptor alpha (Pdgfrα)/Sca1 (PαS) (*Morikawa et al., 2009*), Nestin (*Méndez-Ferrer et al., 2010*), Osterix (*Mizoguchi et al., 2014*), c-x-c motif chemokine 12 (Cxcl12) (*Omatsu et al., 2010*), integrinαV/OX-2 membrane glycoprotein (Cd200) (*Chan et al., 2015*), and early B cell factor 3 (Ebf3) (*Seike et al., 2018*). However, our data suggested that among them, only Sca1 preferentially marks cluster 1. *Pdgfra, Itgav, Gli1,* and *Prrx1* are broadly expressed across all mesenchymal lineage cells; *Sp7 (Osterix)* and *Cd200* are expressed at a higher level in LMPs and LCPs than in EMP; *Grem1, Lepr, Cxcl12* and *Ebf3* are highly expressed in adipocytes but also have expression in other mesenchymal cells (*Figure 1—figure supplement 5A*); *Nestin* and *Mx1* are mainly expressed in endothelial/mural cells and hematopoietic cells, respectively (*Figure 1—figure supplement 5B*). Their expression patterns were further confirmed by UMAP (*Figure 1—figure supplement 5C, D*). Hence, the in vivo EMPs we computationally identified here are similar to the previously proposed PαS cells but different from other proposed MSCs. PαS cells have been shown to display mesenchymal progenitor properties in culture and after transplantation (*Morikawa et al., 2009*).

## The osteogenic and adipogenic lineage commitment of bone marrow mesenchymal progenitors

Positioning individual cells along a linear pseudotimeline with EMPs as the root revealed transcription factors (TFs) differentially expressed after the branch point of osteogenic and adipogenic lineages (*Figure 2A*). Consistent with its longer differentiation pseudotime, adipogenic differentiation required many more unique TFs than osteogenic differentiation, including master regulators *Pparg* and *Cebpa*, several known TFs related to adipogenesis, such as *Klf2, Ebf1,* and *Ebf2* (*Lee et al., 2019*), as well as many novel ones. Surprisingly, this analysis only revealed master osteogenic regulator *Sp7* but not *Runx2*. Indeed, the expression of *Runx2* started from IMPs and stayed constant during osteogenic differentiation (*Figure 1C*). Our observation that adipogenic differentiation is accompanied with more changes in TFs than osteogenic differentiation is consistent with a recent analysis of in vitro adipogenic and osteogenic differentiation of human MSC-TERT4 cells (*Rauch et al., 2019*).

Analyzing differentiated expressed genes (DEGs) revealed that up-regulated genes are distinct for each lineage whereas there is considerable overlap between down-regulated genes (*Figure 2B*). GO term and KEGG analyses of DEGs revealed unique and common features of osteogenic and adipogenic differentiation processes (*Figure 2C*). Some pathways, such as PI3K-Akt, TGFβ, and FoxO signaling pathways, were altered in both lineages, implying their general role in regulating mesenchymal differentiation. Accompanied by downregulation of translation, ribosomal genes were downregulated in both osteocytes and adipocytes, confirming that they are terminally differentiated, highly specialized cells. The unique features of osteogenic differentiation include extracellular matrix (ECM) organization, ECM-receptor interaction, axon guidance, actin cytoskeleton, and biomineral tissue development, all of which reflect their bone building function. Interestingly, the unique features of adipogenic differentiation include cytokine-cytokine receptor interaction, immune system process, circadian rhythm, osteoclast differentiation, TNF, HIF-1, chemokines signaling pathways, among others. Considering that adipocytes reside in a hematopoietic environment with affluent blood vessels, these features suggest an important regulatory role of adipocytes on its marrow environment.

To validate the lineage differentiation routes predicted by our Seurat and Monocle data, we chose two genes, *Dmp1* and *Acta2 (αSMA)*, for further investigation. DMP1 was previously reported as an osteocyte marker (*Lu et al., 2007*). Our sequencing analysis revealed that its expression is turned on in LCPs mostly in osteogenic differentiation route (*Figure 2D,E*). In 3-month-old *Dmp1-Cre Rosa26 <lsl-tdTomato>* (Dmp1:Td) mice, Td labeled almost all osteoblasts and osteocytes, some stromal cells and a few Perilipin[+] adipocytes (6 out of 92 adipocytes counted, n = 3 mice, 6.1%, *Figure 2F*), which is similar to a previously reported Dmp1-Cre targeting pattern (*Lim et al., 2016*). αSMA was previously reported as an osteoprogenitor marker (*Kalajzic et al., 2008*). We found that it is a marker for the LMP cluster (*Figure 2G,H*). In *Acta2-CreER Rosa26 <lsl-tdTomato>*

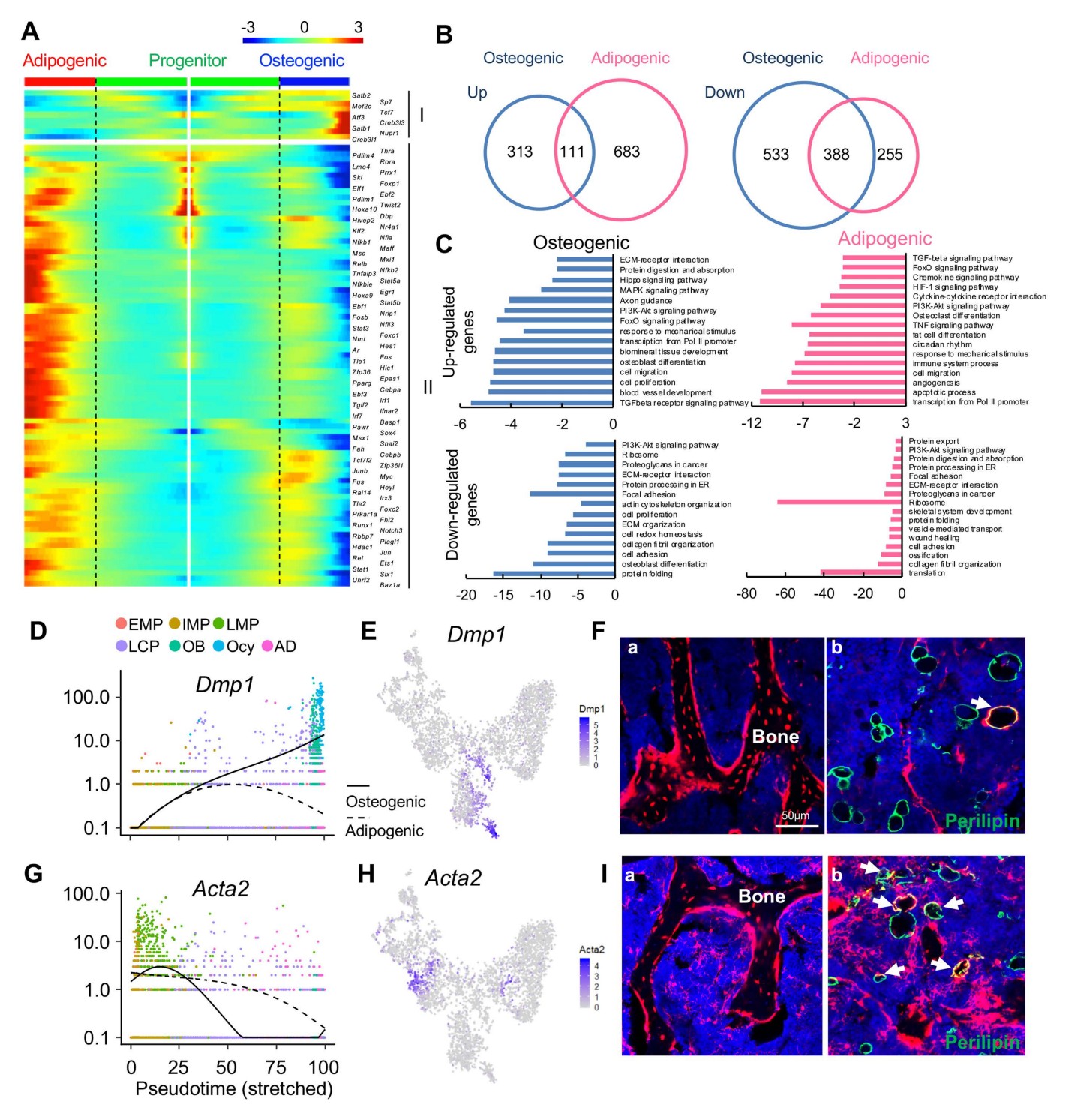

**Figure 2.** The bifurcated osteo- and adipo-lineage differentiation routes of in vivo bone marrow mesenchymal progenitors. (**A**) Pseudotemporal depiction of differentially expressed transcription factors (TFs) starting from branching point (dashed lines) toward osteo- (right) and adipo- (left) lineage differentiation. Group I and II contain TFs that are highly up-regulated during osteogenic and adipogenic differentiation routes, respectively. Color bar indicates the gene expression level. (**B**) Differentially regulated (up-regulated and down-regulated) genes during osteogenic and adipogenic differentiation are counted. (**C**) GO term and KEGG pathway analyses of genes up-regulated and down-regulated during osteogenic and adipogenic differentiation. Note that some pathways, such as osteoblast differentiation, are identified by both up-regulated and down-regulated genes. This is due to the fact that a set of genes in a pathway are up-regulated while another set of genes in the same pathway are down-regulated. (**D**) Expression of

*Figure 2 continued on next page*

Figure 2 continued

*Dmp1* goes from the progenitor state and bifurcating into osteogenic or adipogenic branches with respect to pseudotime coordinates. (E) The tSNE plot predicts *Dmp1* expression in osteoblasts, osteocytes, and a portion of LCPs. (F) In 3-month-old Dmp1:Td mice, Td labels osteoblasts, osteocytes (a), and only a few adipocytes (b, arrows). (G) Expression of *αSMA* (*Acta2*) goes from the progenitor state and bifurcating into osteogenic or adipogenic branches with respect to pseudotime coordinates. (H) The tSNE plot predicts *αSMA* expression in LMPs. (I) In 4-month-old Acta2ER:Td mice with Tamoxifen injections at 1 month of age, Td labels osteoblasts, osteocytes (a) and many adipocytes (b, arrows).

(Acta2ER:Td) mice at 3 months after Tamoxifen injections, Td labeled many osteoblasts, osteocytes (185 out of 256 osteocytes counted, n = 3 mice, 68.2%), and bone marrow Perilipin$^+$ adipocytes (42 out of 124 adipocytes counted, n = 3 mice, 33.7%, *Figure 2I*), confirming that αSMA labels mesenchymal progenitors before bifurcated differentiation.

## Age-dependent changes in bone marrow mesenchymal subpopulations

Mouse bone marrow mesenchymal progenitor pool shrinks drastically over time. Indeed, CFU-F frequency of endosteal bone marrow harvested from 16-month-old *Col2:Td* mice decreased by 73% compared to that from 1-month-old mice (*Figure 3A*). Similar to adolescent mice, almost all CFU-F colonies from adult and aging mice were Td$^+$ (*Figure 3A*) and top 1% Td$^+$ cells sorted from endosteal bone marrow contained almost all CFU-Fs (100% Td$^+$) in unsorted cells (*Figure 3—figure supplement 1A, B*). These data support the use of a similar scRNA-seq approach on adult and aging bone marrow mesenchymal lineage cells.

After quality control, we obtained 4502 cells with 19 clusters in a 3 month dataset and 8823 cells with 16 clusters in a 16 month dataset (*Figure 3—figure supplement 1C*). Among them, 2510 (2040 genes/cell and 7243 UMIs/cell) and 7354 (2671 genes/cell and 9660 UMIs/cell) were mesenchymal lineage cells in 3 and 16 month dataset, respectively. Unsupervised clustering of cells in both datasets using tSNE and UMAP yielded similar clustering patterns (*Figure 3B*, *Figure 3—figure supplement 1D*) and cluster markers (*Figure 3C*) as 1 month dataset. Notably, IMP cluster was not detected in 16 month dataset. After removing chondrocytes, pseudotime trajectories using either Monocle or Slingshot again put EMPs at one end and osteocytes and adipocytes at the other two ends (*Figure 3D*).

Merging all age datasets generated 6 clusters: EMP, LMP, osteoblast, osteocyte, LCP, and adipocyte (*Figure 3E*). It is conceivable that the mesenchymal progenitor pool with bi-lineage differentiation ability was drastically shrunk in the aging sample while the adipocyte population was greatly expanded (*Figure 3F-G*), which is consistent with our CFU-F results. Pseudotime trajectory analyses again revealed a similar Y shape curve as 1 month dataset (*Figure 3—figure supplement 2A*). Interestingly, while EMP and LMPs in 1 and 3 month datasets were more centered at the starting point of pseudotime, those cells in 16 month dataset shifted toward differentiated status, particularly the adipocyte end (*Figure 3—figure supplement 2B, C*). We further examined the expression pattern of adipocyte markers in each cell cluster among different age groups (*Figure 3—figure supplement 2D*). Some of them, such as *Cebpa* and *Lpl*, were expressed at higher levels in most mesenchymal subpopulations, particularly EMP and LMP, in 16 month dataset comparing to 1 and 3 month datasets, suggesting an adipocytic drift during aging. Lepr was previously described as a marker labeling adult MSCs but not young MSCs (*Zhou et al., 2014*). Here we observed that Lepr expression is much higher in 16 month EMPs than in 1 month EMPs, likely providing an explanation for why *Lepr-Cre* labels adult MSCs but not young MSCs in mouse bone marrow (*Zhou et al., 2014*).

In summary, we conclude that the same pool of EMPs with the same hierarchy differentiation pattern is responsible for bone formation by mesenchymal lineage cells at adolescent, adult, and aging stages. During aging, EMPs are not only reduced in numbers but also drifted toward more adipogenic status, which might further account for the loss of progenitor activity.

## Discovery of a novel adipogenic lineage cell population in bone

The conventional view of bone marrow adipocytes is that they are Perilipin$^+$ cells with a single large lipid droplet (*Horowitz et al., 2017*). However, due to their large size and buoyance, those cells cannot be captured by pelleting and cell sorting. Furthermore, they are rare in the adolescent mouse bones. Therefore, we were surprised to identify a large cell cluster in young mice that express

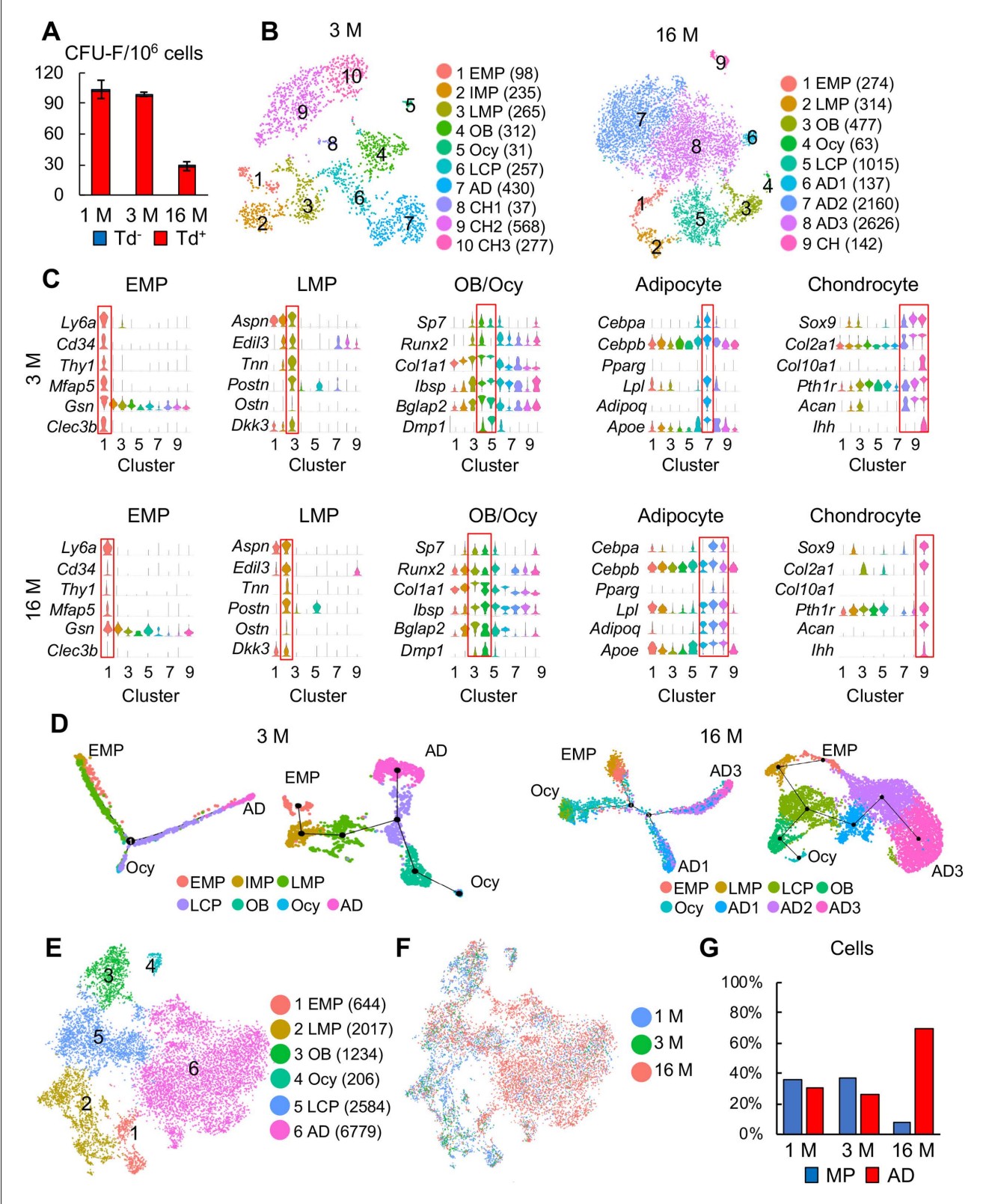

**Figure 3.** Large scale scRNA-seq analyses of bone marrow mesenchymal lineage cells from 3- and 16-month-old Col2:Td mice confirm the same in vivo mesenchymal subpopulations as 1-month-old mice. (**A**) CFU-F assays of endosteal bone marrow cells from 1-, 3-, and 16-month-old Col2:Td mice. n = 3 mice/group. Td+ and Td- CFU-F colonies are quantified. (**B**) The tSNE plots of Td+ mesenchymal lineage cells isolated from endosteal bone marrow of *Figure 3 continued on next page*

Figure 3 continued

3- and 16-month-old Col2:Td mice (n = 3 mice/group). Cell numbers are listed in parenthesis next to cluster names. OB: osteoblast; Ocy: osteocyte; LCP: lineage committed progenitor; AD: adipocyte; CH: chondrocyte. (C) Violin plots of marker gene expression for indicated clusters in 3 and 16 month datasets. (D) Monocle (left) and Slingshot (right) trajectory plots of bone marrow mesenchymal lineage cells (excluding chondrocytes) in 3 and 16 month datasets. Cells are labeled according to their Seurat clusters. (E) An integrated tSNE plot of 1, 3, and 16 month datasets shows the clustering pattern. (F) An integrated tSNE plot of 1, 3, and 16 month datasets shows the distribution of each age group. (G) The percentages of mesenchymal progenitors before lineage commitment and adipocytes (ADs) within bone marrow mesenchymal lineage cells are quantified in each age group based on tSNE distribution.

The online version of this article includes the following figure supplement(s) for figure 3:

**Figure supplement 1.** Large scale scRNA-seq analysis of top 1% Td⁺ cells from the endosteal bone marrow of 3- and 16-month-old Col2:Td mice.
**Figure supplement 2.** Pseudotime trajectory analysis of bone marrow mesenchymal lineage cells from all age groups.

---

common adipocyte markers, including *Pparg, Cebpa, Adipoq, Apoe,* and *Lpl* (**Figure 4—figure supplement 1A, C**). However, they do not express *Plin1,* a gene encoding lipid droplet coating protein, and they express *Fabp4,* a gene encoding a fatty acid binding protein, at a very low level, implying that they do not contain lipid droplets. In addition, these cells express a variety of other group defining genes, such as *Lepr, Cxcl12, Il1rn, Serpina3g, Kng1, Kng2, Agt, Esm1,* and *Gdpd2* (**Figure 4— figure supplement 1B, D**). qRT-PCR analysis demonstrated that those genes are indeed up-regulated during adipogenic differentiation of mesenchymal progenitors in culture (**Figure 4—figure supplement 2**), supporting that cluster 7 in 1 month dataset represents adipose lineage cells.

To validate this population, we constructed adipocyte-specific *Adipoq-Cre* reporter mice: *Adipoq:Td* mice with or without *Col1a1-GFP* that labels osteoblasts. In line with the above sequencing data, there were many Td⁺ cells, existing either as CD45⁻ stromal cells or pericytes, in the bone marrow of newborn pups (**Figure 4—figure supplement 3**) when long bone undergoes rapid bone formation. At 1 month of age, Td labeled all Perilipin⁺ adipocytes, CD45⁻ stromal cells with a reticular shape, and pericytes, but not osteoblasts, osteocytes, growth plate or articular chondrocytes (**Figure 4A**). The majority of Td⁺ cells did not harbor lipid droplets (**Figures 4B**, 1995 out of 2000 cells counted, n = 6 mice, 99.8%) and none of them incorporated EdU (**Figures 4C**, 0 out of 2060 cells counted, n = 3 mice, 0%). After isolated from bone marrow, Td⁺ cells attached to the culture dish but did not form CFU-F colonies (**Figures 4D**, 0 out of 100 colonies counted, n = 3 mice, 0%). In freshly isolated endosteal bone marrow, Td⁺ cells constituted about 18% of CD45⁻Ter119⁻ cells. During culture, the percentage of Td⁺ cells decreased progressively after cell passage (P0 when CFU-F colonies became confluent: 2.03%; P1: 0.93%: P2: 0.31%, by flow cytometry analysis), suggesting that those cells have much less proliferative ability if any compared to mesenchymal progenitors. Moreover, while freshly sorted Td⁺ cells from bone marrow of Col2:Td mice formed bone with a hematopoietic compartment after transplanted under mouse kidney capsule, freshly sorted Td⁺ cells from Adipoq:Td bone did not form bone-like structure (**Figure 4E**), demonstrating that they are not mesenchymal progenitors. This is consistent with the notion that *Adipoq-Cre* does not label adipoprogenitors in peripheral fat depots (**Liu et al., 2017**). Indeed, the stromal vascular fraction (SVF) of visceral fat from Adipoq:Td mice contained no Td⁺ CFU-F colonies (data not shown).

To delineate the relationship among mesenchymal progenitors, non-lipid-laden Perilipin⁻Td⁺ cells, and lipid-laden Perilipin⁺Td⁺ cells, we subjected confluent mesenchymal progenitors from Adipoq:Td mice, which are Td⁻, to adipogenic differentiation. Interestingly, Td⁻ cells (day 0) became Td⁺ cells (days 1 and 2) with no lipid droplets and then evolved into Td⁺ cells with lipid accumulation (days 3–5, **Figure 4F**). For fate mapping in vivo, we generated *Adipoq-CreER Rosa26 <lsl-tdTomato>* (AdipoqER:Td) mice. Tamoxifen injections at P14-16 induced many Td⁺ cells in bone marrow at P23 with a similar distribution pattern as Td⁺ cells in Adipoq:Td mice (**Figure 4—figure supplement 4**). All CFU-Fs from their bone marrow were Td⁻ and only a few Td⁺ cells attached to the dish without further proliferation (**Figure 4G**). AdipoqER:Td mice receiving Tamoxifen injections at P6-7, when no Perilipin⁺ adipocytes can be detected in the proximal tibia, displayed Perilipin⁺Td⁺ adipocytes at 1 month of age (**Figure 4H**), suggesting that Perilipin⁻Td⁺ cells become bone marrow adipocytes in vivo. Taken together, our data clearly demonstrate that non-lipid-laden, *Adipoq-Cre* labeled cells constitute a mesenchymal subpopulation situated after mesenchymal progenitors and before classic lipid-laden adipocytes (LiLAs) along the adipogenic differentiation route of bone marrow EMPs. As a

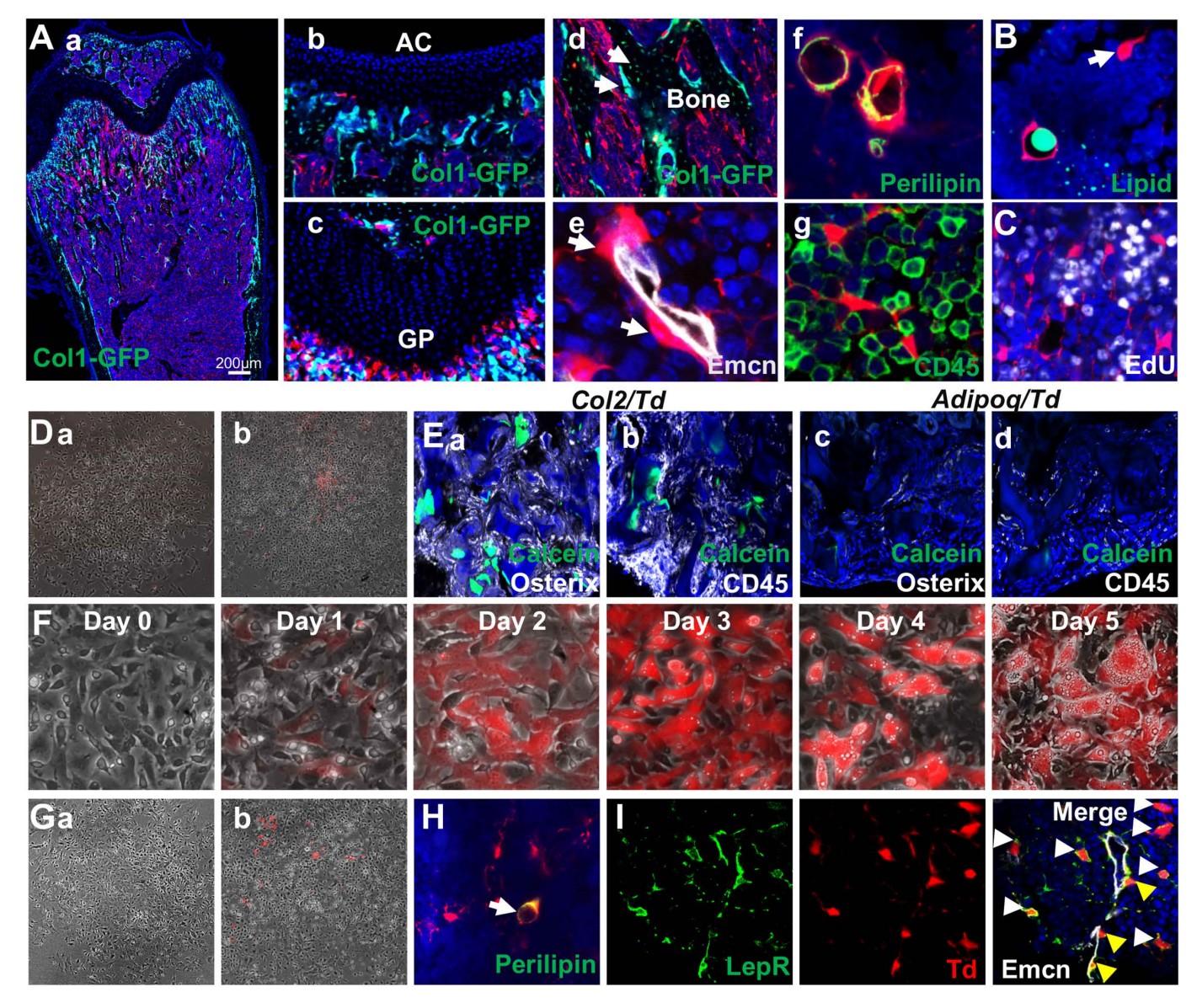

**Figure 4.** Mouse bone marrow contains abundant non-proliferative adipocyte precursors. (A) Representative fluorescent images of 1-month-old Adipoq:Td:Col1a1-GFP (a–d) or Adipoq:Td (f–g) femur reveal many bone marrow Td+ cells. (a) A low magnification image of a distal femur. (b–g) At a high magnification, Td does not label chondrocytes in articular cartilage (b) and growth plate (c), osteoblasts and osteocytes (arrows, d) but labels pericytes (arrows, e), Perilipin+ adipocytes (f), and CD45- stromal cells (g). (B) According to BODIPY lipid staining, those Td+ stromal cells in a reticular shape (arrow) have no lipid accumulation. (C) In vivo EdU injection reveals that Td+ cells in Adipoq:Td mice do not proliferate. (D) CFU-F assay of bone marrow cells from Adipoq:Td mice shows that all CFU-F colonies are made of Td- cells (a). (b) Some Td+ cells do attach to the dish but have minimum proliferation ability. n = 5 mice. (E) Endosteal bone marrow Td+ cells from 1-month-old Col2:Td mice (5 out of 5 transplants), but not Adipoq:Td mice (0 out of 3 transplants) form bone-like structure after transplanted under the kidney capsule. Representative fluorescent images of transplants were shown here. Osterix: osteoblasts; CD45: hematopoietic cells; Calcein: new bone surface. (F) In vitro adipogenic differentiation assay demonstrates that non-lipid-laden adipocytes exist as an intermediate state between mesenchymal progenitors and lipid-laden adipocytes. Mesenchymal progenitors, which are Td-, were obtained from culturing endosteal bone marrow cells from 1-month-old Adipoq:Td mice. Upon confluency, cells were cultured in adipogenic differentiation medium. The same area was imaged daily by inverted fluorescence microcopy. (G) CFU-F assay of bone marrow cells from P23 AdipoqER:Td mice (Tamoxifen injections at P14-16) shows that all CFU-F colonies are made of Td- cells (a). (b) Some Td+ cells do attach to the dish but have minimum proliferation ability. n = 3 mice. (H) Immunofluorescence staining shows that Perilipin+bone marrow adipocytes are derived from non-lipid-laden adipocytes in 1-month-old AdipoqER:Td mice (Tamoxifen injections at P6, 7). (I) Representative images of co-localization of Td and Lepr in stromal (white arrows) and pericytes (yellow arrows) in bone marrow of 1-month-old Adipoq:Td mice.

*Figure 4 continued on next page*

*Figure 4 continued*

The online version of this article includes the following figure supplement(s) for figure 4:

**Figure supplement 1.** Large scale scRNA-seq data predict a large cluster of adipocytes in the bone marrow of adolescent mice.
**Figure supplement 2.** The adipocyte markers predicted by sequencing data are validated by in vitro adipogenic differentiation assay.
**Figure supplement 3.** New born Adipoq:Td mice (P5) have abundant Td$^+$ cells in the bone marrow.
**Figure supplement 4.** Young AdipoqER:Td mice have abundant Td$^+$ cells in bone marrow.

further validation that *Adipoq-Cre* labeled cells are cells in cluster 7 of 1 month dataset, we stained sections with Lepr, a cluster 7 marker. As expected, almost all Td$^+$ cells are Lepr$^+$ as well (*Figure 4I*).

Compared to conventional adipose depots, we did not detect the expression of *Lep* (white adipocyte marker), *Ucp1* (brown adipocyte marker), *Tnfrsf9* (beige adipocyte marker) in cells of adipocyte cluster based on our sequencing data. Other adipose depot markers, such as *Hoxc8*, *Hoxc9* (white), *Cidea*, *Cox7a1*, *Zic1* (brown), *Cited1*, *Shox2*, *Tbx1* (beige) (*Cheng et al., 2018*; *Wu et al., 2012*), were expressed either at very low level or ubiquitously among most mesenchymal lineage cells. Overall, our results indicate that bone marrow contains a unique, large population of adipogenic lineage cells that have no lipid storage.

## A 3D network in bone marrow formed by *Adipoq*-labeled stromal cells and pericytes

In the bone marrow of young Adipoq:Td mice, Perilipin$^-$Td$^+$ cells far exceeded Perilipin$^+$Td$^+$ adipocytes in number. Cell counting in tibial sections revealed approximate 400:1 and 1000:1 ratios of Perilipin$^-$Td$^+$ cells to Perilipin$^+$Td$^+$ in the metaphyseal and diaphyseal bone marrow, respectively (n = 3 mice). Morphologically, Perilipin$^-$Td$^+$ cells exist as stromal cells and pericytes. PDGFRβ is a general pericyte marker (*Crisan et al., 2008*) and Laminin is a component of basement membrane secreted by both pericytes and endothelial cells (*Yousif et al., 2013*). Our scRNA-seq data showed that both *Pdgfrb* and *Lamb1* are expressed at relatively higher levels in the adipocyte cluster among all mesenchymal clusters (*Figure 5A*, *Figure 5—figure supplement 1*). Pericytes identified by PDGFRβ or Laminin staining in a peri-capillary location were Td$^+$ (850 out of 850 cells counted, n = 3 mice, 100%, *Figure 5B*), indicating that *Adipoq* marks pericytes surrounding bone marrow capillaries. To further confirm their pericyte nature, we mixed sorted Td$^+$ cells from Adipoq:Td bone marrow with endothelial progenitors and conducted vasculogenic tube assays. Strikingly, most Td$^+$ cells co-localized with newly formed tubes (*Figure 5C*), suggesting that they function as pericytes. Consistent with this, the adipocyte cluster express many angiogenic factors, such as *Vegfa*, *Vegfc*, *Angpt4*, *Rspo3* etc (*Figure 5D*, *Figure 5—figure supplement 1*). Gene profiling of fractionated Td$^+$ cells from Adipoq:Td bone marrow confirmed the remarkably high expression of these angiogenic factors as well as adipocyte markers and other newly identified markers (*Figure 5E*). Bone marrow pericytes seem to be unique because we did not observe Td$^+$ pericytes in other tissues of Adipoq:Td mice, such as muscle, liver, lung, brain, heart, and kidney (*Figure 5—figure supplement 2*).

Interestingly, when we scanned 50 µm-thick sections of Adipoq:Td long bones, we observed a striking 3D network made of cell processes from Td$^+$ cells that were ubiquitously distributed inside bone marrow (*Figure 5Fa, b*, *Figure 5—video 1*). In sharp contrast to round hematopoietic cells, Td$^+$ stromal cells and pericytes possessed an average of 6.1 ± 0.2 cell processes (52 cells counted, n = 3 mice), which were covered by cell membrane proteins, such as PDGFRβ and gap junction Connexin 43 (*Figure 5Fc*, d, *Figure 5—videos 2* and *3*). These processes were reminiscent of osteocytic canaliculi structure, although they are much longer and more disorientated. With these processes, Td$^+$ stromal cells and pericytes were morphologically similar (*Figure 5Fe*, *Figure 5—video 4*). They all extended cell processes into the marrow, making numerous connections amongst themselves, around endothelial walls (*Figure 5Fe*), with hematopoietic cells (*Figure 5Ff*), and with bone surface (*Figure 5Fg*). In LiLAs, cell processes were only detected at a small portion of cell surface next to cytoplasm and nuclei but not at the most part of cell surface next to the lipid droplet (*Figure 5Fh*, *Figure 5—video 5*). The vast 3D structure of Td$^+$ cells implies a unique communication role.

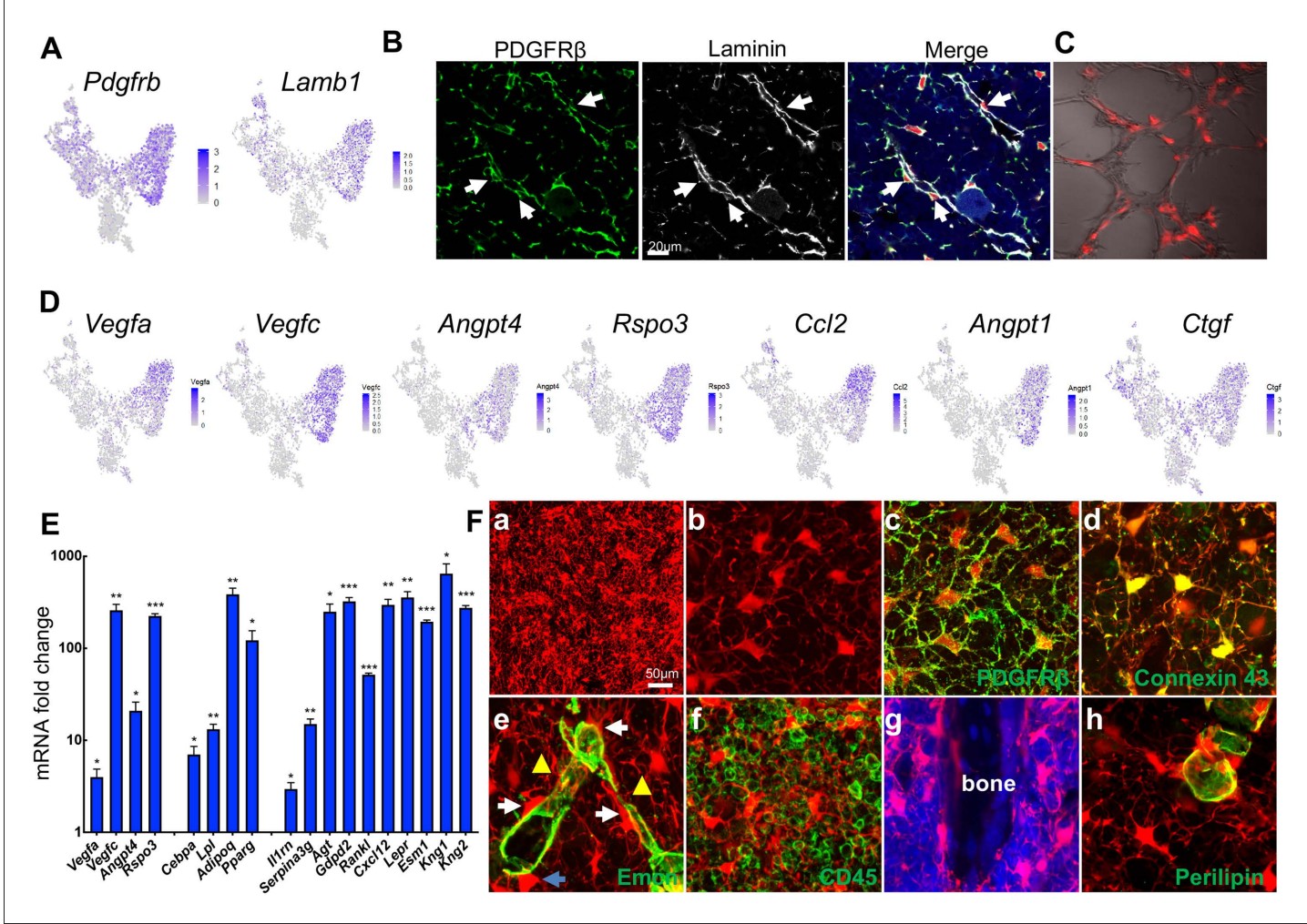

**Figure 5.** Non-lipid-laden Td+ cells in Adipoq:Td mice form a vast 3D network through their cell processes ubiquitously distributed inside the bone marrow. (**A**) The expression patterns of *Pdgfrb* and *Lamb1* are shown in tSNE plots. (**B**) Immunofluorescence staining reveals that all PDGFRβ+ and Laminin+ cells with a pericyte morphology are Td+ (pointed by arrows). (**C**) Td+ cells sorted from Adipoq:Td bone marrow act as pericytes after cocultured with bone marrow endothelial cells undergoing tube formation assay. (**D**) The expression patterns of angiogenic factors are shown in tSNE plots. (**E**) qRT-PCR analyses comparing mRNA levels in Td+ versus Td- cells (set as 1) sorted from bone marrow of 1-month-old Adipoq:Td mice confirm that *Adipoq-Cre* labeled cells highly express angiogenic factors, known adipocyte markers, and novel markers suggested by sequencing data. *: p<0.05; **: p<0.01; ***: p<0.001 Td+ vs Td-, n = 3 mice. (**F**) A 3D network in bone marrow formed by Td+ stromal cells and pericytes in 1-month-old Adipoq:Td mice. (**a**) A low magnification image reveals the network of Td+ cells in the femoral bone marrow. (**b**) Td+ cell bodies and their processes are more obvious in a high magnification image. See *Figure 5—video 1* for a 3D structure. (**c**) Those Td+ cells have PDGFRβ staining all over their cell processes. See *Figure 5—video 2* for a 3D structure. (**d**) They also have punctuated Connexin 43 staining in their process, indicating cell-cell communication. See *Figure 5—video 3* for a 3D structure. (**e**) Td+ pericytes have cell processes. Some processes protrude into bone marrow just like those of Td+ stromal cells and some of them wrap around the vessel wall. Similarly, Td+ stromal cells also extend their processes toward vessels either wrapping around the vessel wall (yellow triangles) or contacting the processes from pericytes. White arrows point to cells with a typical pericyte morphology. A blue arrow points to a Td+ cell sitting on a vessel with a stromal cell shape. Therefore, considering the presence of cell processes and connections, Td+ stromal cells and pericytes are indeed very similar. See *Figure 5—video 4* for a 3D structure. (**f**) Cell processes from Td+ cells touch almost every CD45+ hematopoietic cells inside bone marrow. (**g**) They also reach trabecular bone surface. (**h**) On the contrary, most cell surface of Perilipin+Td+ adipocytes are smooth and round. See *Figure 5—video 5* for a 3D structure.

The online version of this article includes the following video and figure supplement(s) for figure 5:

**Figure supplement 1.** The expression patterns of pericyte markers and angiogenesis factors in UMAP plots.

**Figure supplement 2.** Td+ pericytes are unique to the bone marrow of Adipoq:Td mice.

**Figure 5—video 1.** Confocal fluorescence image of Td+ cells in bone marrow of Adipoq:Td mice to show its 3D network structure made of cell processes protruding from cell bodies.

**Figure 5—video 2.** Confocal fluorescence image of Td+ cells with PDGFRβ staining (green) in bone marrow of Adipoq:Td mice.

*Figure 5 continued on next page*

*Figure 5 continued*

https://elifesciences.org/articles/54695#fig5video2

**Figure 5—video 3.** Confocal fluorescence image of Td$^+$ cells with Connexin 43 staining (green, shown as dots on cell processes) in bone marrow of Adipoq:Td mice.

https://elifesciences.org/articles/54695#fig5video3

**Figure 5—video 4.** Confocal fluorescence image of Td$^+$ cells with Emcn staining (green, vessel) in bone marrow of Adipoq:Td mice.

https://elifesciences.org/articles/54695#fig5video4

**Figure 5—video 5.** Confocal fluorescence image of Td$^+$ cells with Perilipin staining (green) in bone marrow of Adipoq:Td mice.

https://elifesciences.org/articles/54695#fig5video5

## The critical function of *Adipoq*-labeled cells in bone

We next sought to study the function of these Td$^+$ cells by ablating them in 1-month-old *Adipoq-Cre Rosa26 <lsl-tdTomato>DTR* (Adipoq:Td:DTR) mice via diphtheria toxin (DT) injections. After two weeks, DT-treated mice displayed pale bones as compared to vehicle-treated mice (*Figure 6A*). In bone marrow Perilipin$^+$Td$^+$ and Perilipin$^-$Td$^+$ cells were greatly reduced by 95% and 88%, respectively (*Figure 6B,C*). Bone marrow vasculature underwent severe pathological changes, becoming dilated and distorted (*Figure 6D,E*). Quantification revealed 2.3- and 1.9-fold increases in vessel diameter and vessel area, respectively, and a 36% decrease in vessel density (*Figure 6F*). Additionally, the number of Emcn$^+$CD31$^+$ endothelial cells decreased by 45% after DT treatment (*Figure 6G*). In bone marrow, *Adipoq-Cre* labeled a similar density of pericytes as *Col2-Cre* (0.0172 ± 0.0019 vs 0.0174 ± 0.0028 pericytes/μm vessel length, n = 3 mice/group). Strikingly, DT injections drastically decreased Td$^+$ pericytes by 90% to 0.0018 ± 0.0005 pericytes/μm vessel length (n = 3 mice/group, *Figure 6E*). Hence, one role of these non-lipid laden Td$^+$ cells is to maintain normal bone marrow vessels.

The color change in bone after cell ablation was due to de novo trabecular bone formation throughout the marrow cavity, especially in the diaphysis that is normally devoid of any trabeculae (*Figure 7A–C*). MicroCT quantification of newly formed bone in the diaphyseal region revealed that it resembles metaphyseal trabecular bone in normal mice (*Figure 7D*). Numerous Osterix$^+$ osteoblasts were observed inside the diaphyseal bone marrow (*Figure 7E*), suggesting that osteogenic differentiation is rapidly activated. The fact that nearly all osteoblasts are Td$^-$ further confirmed that *Adipoq-Cre* rarely labels mesenchymal progenitors that generate osteoblasts. Furthermore, DT injections drastically reduced CFU-F frequency from bone marrow (*Figure 7F*), suggesting a direct or indirect effect of Td$^+$ cells on progenitors. The new bone formation along the endosteal surface of cortical bone was mostly woven bone with misaligned collagen fibrils (*Figure 7G*). This resulted in increased cortical thickness and area, reduced endosteal perimeter, and low tissue mineral density (*Figure 7H*).

After DT treatment, the overall bone marrow cellularity was significantly decreased (*Figure 7—figure supplement 1A*). However, the hematopoietic components, including hematopoietic stem/progenitors cells and lineage specific cells, as well as peripheral blood components, remained largely unaltered (*Figure 7—figure supplement 1B, C*). Therefore, the reduced cellularity is likely caused by the smaller marrow cavity secondary to the new bone formation.

Non-lipid-laden Td$^+$ cells were also abundant in the vertebrae of adolescent mice (*Figure 7—figure supplement 2A*). After cell ablation in Adipoq:Td:DTR mice, bone marrow vasculature was similarly distorted (*Figure 7—figure supplement 2B*), and osteogenic cells were similarly increased (*Figure 7—figure supplement 2C*), leading to elevated trabecular bone mass (*Figure 7—figure supplement 2D, E*). Note that DT injection alone did not alter bone structure in long bones or vertebrae of Adipoq:Td mice (data not shown).

*Adipoq-Cre* also targets peripheral fat depots. A recent study on 'fatless' mice (*Adipoq-Cre DTA*) reported that circulating adiponection and leptin released by subcutaneous adipocytes negatively regulate bone formation (*Zou et al., 2019*). To investigate whether a similar mechanism occurs in our cell ablation model, we transplanted *WT* subcutaneous fat to Adipoq:Td:DTR mice and performed DT injections 2 weeks later. While DT injections greatly increased blood glucose level, this increase was abolished by fat transplantation (*Figure 7—figure supplement 3A*), suggesting a successful restoration of peripheral fat in DT-treated mice. Interestingly, in our mouse model, fat

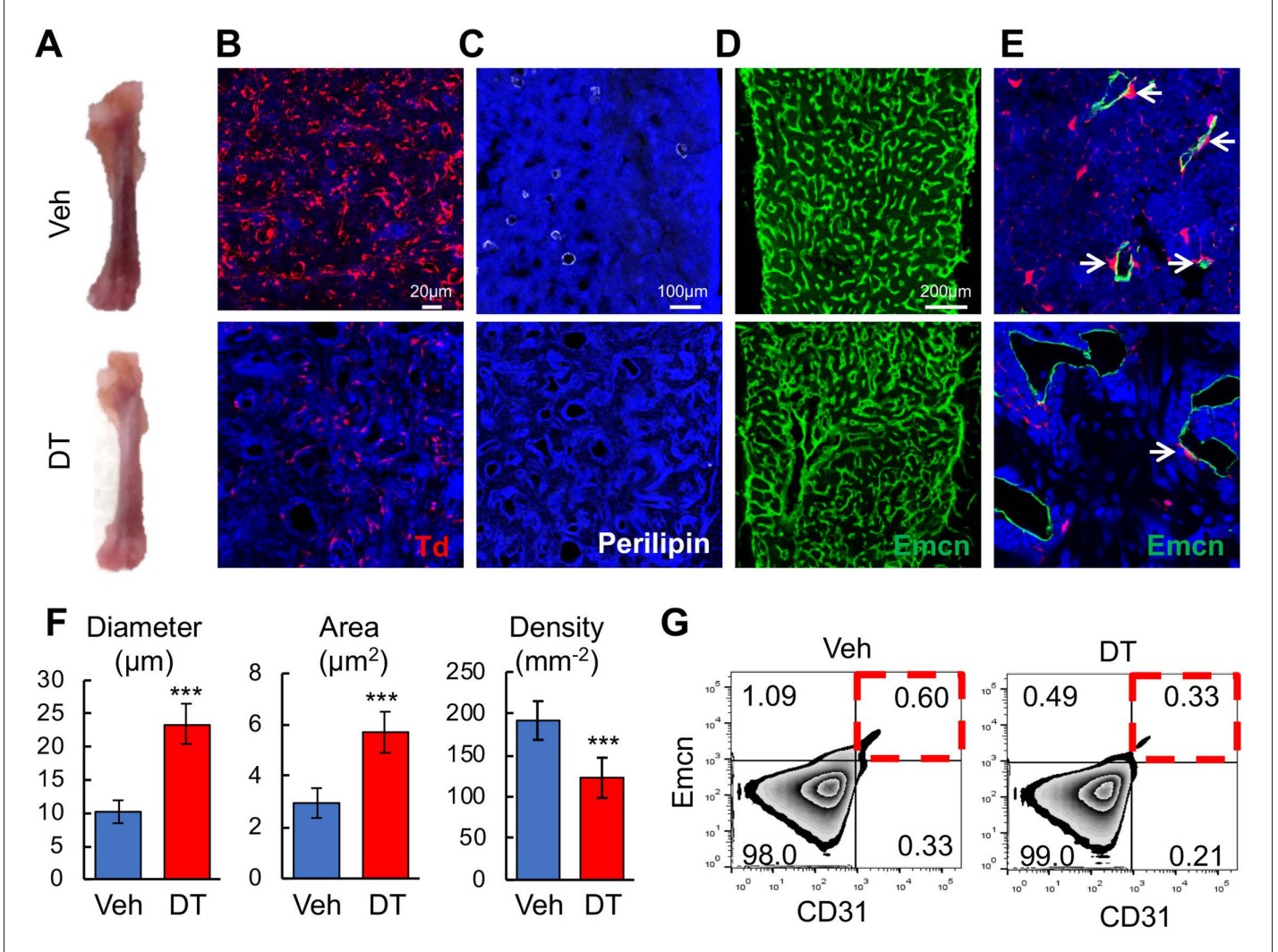

**Figure 6.** Bone marrow Adipoq+ cells are critical for supporting marrow vasculature. (A) Ablation of *Adipoq-Cre* labeled cells changes the color of long bones from red to pink. One-month-old Adipoq:Td:DTR mice received either veh or DT injections (50 µg/kg) every other day for 2 weeks and long bones were harvested for analysis. (B) Inside the long bone, Td+ cells were significantly reduced by DT injections. (C) Perilipin+ adipocytes were also diminished by DT injections. (D) Fluorescent images of vessel staining in femur at a low magnification revealed abnormal bone marrow vessel structure after DT injections. (E) Fluorescent images of vessel staining in femur at a high magnification showed that vessels were dilated coinciding with the depletion of Td+ pericytes (arrows) after DT injections. (F) Vessel diameter, area, and density are quantified in the central bone marrow. n = 3 mice. ***: p<0.001 DT vs veh. (G) Flow cytometric analysis of endothelial cells (Emcn+CD31+) in bone marrow after DT injections.

transplantation did not appear to affect DT-induced vessel damage and de novo formation in the diaphysis bone marrow (*Figure 7—figure supplement 3B-E*), suggesting that peripheral fat tissues are not involved in the drastic changes in Adipoq:Td:DTR bone after DT injections.

Taken together, our data indicate that these *Adipoq-Cre* labeled, non-lipid-laden cells are a critical bone marrow component regulating their environment, including marrow vasculature and bone formation. Therefore, we name this new type of adipogenic lineage cell as marrow adipogenic lineage precursors (MALPs).

## Discussion

In this study, we computationally delineated the entire in vivo differentiation process of mesenchymal progenitors and defined the mesenchymal lineage hierarchy inside bone marrow (*Figure 8*). Stem cell heterogeneity and plasticity have been recognized in some well-studied mammalian tissue

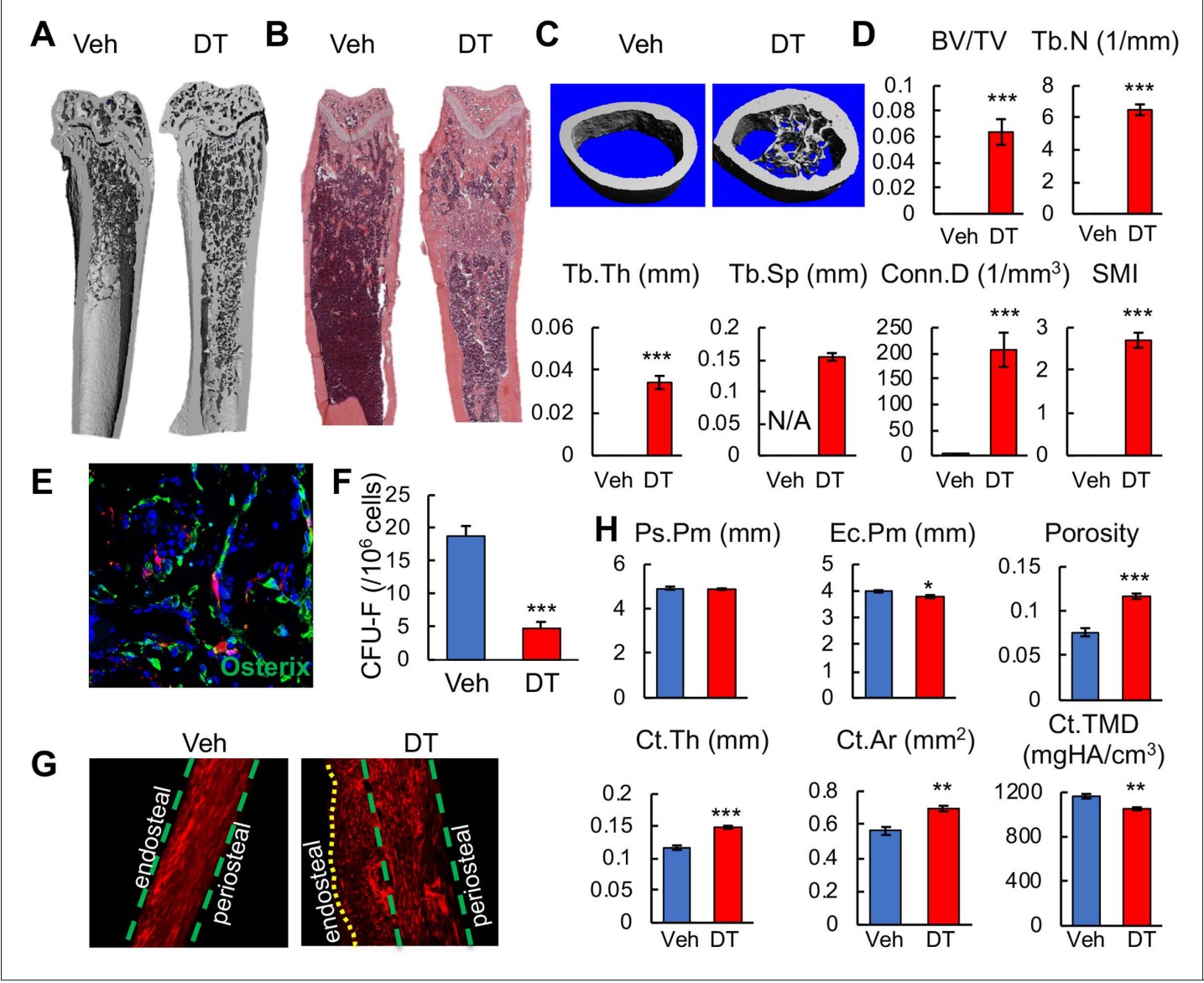

**Figure 7.** Bone marrow Adipoq[+] cells inhibit bone formation. (**A**) 3D μCT images of Adipoq:Td:DTR femurs reveal drastic de novo bone formation in the midshaft region after 2 weeks of DT injections. (**B**) H and E staining of femoral sections. (**C**) 3D reconstructed microCT images of femoral midshaft region. (**D**) MicroCT measurement of trabecular bone structural parameters from the midshaft region. BV/TV: bone volume fraction; Tb.N: trabecular number; Tb.Th: trabecular thickness; Tb.Sp: trabecular separation; Conn.D: connective density; SMI: structural model index. n = 6 mice/group. *: p<0.05; ***: p<0.001 DT vs veh. (**E**) Fluorescent image shows that newly formed osteoblasts (Osterix[+] cells) in the diaphyseal bone marrow are Td[-] cells. (**F**) CFU assay demonstrates a decrease in mesenchymal progenitors in Adipoq:Td:DTR femurs after DT injections. n = 3 mice/group. ***: p<0.001 DT vs veh. (**G**) Second harmonic generation images of femoral sections reveal that collagen fibers in the newly formed bone on the endocortical surface are mostly misaligned in Adipoq:Td:DTR mice after DT injections. (**H**) MicroCT measurement of cortical bone structural parameters from the midshaft region. Ps.Pm: periosteal perimeter; Ec.Pm: endosteal perimeter; Ct.Th: cortical thickness; Ct.Ar: cortical area; Ct.TMD: cortical tissue mineral density. n = 6 mice/group. *: p<0.05; **: p<0.01; ***: p<0.001 DT vs veh.

The online version of this article includes the following figure supplement(s) for figure 7:

**Figure supplement 1.** Ablation of Adipoq[+] cells reduces overall bone marrow (BM) cellularity but has little effect on hematopoietic cells.

**Figure supplement 2.** Ablation of Adipoq[+] cells increases trabecular bone mass in vertebrae.

**Figure supplement 3.** Fat transplantation does not rescue bone phenotypes induced by ablation of Adipoq[+] cells.

stem cells (**Tang, 2012**). However, in the past, owing to a lack of discerning investigative tools, we purposely ignored these features of mesenchymal lineage cells by simply referring to all progenitors as MSCs or mesenchymal progenitors and searching for one or a set of marker(s) to cover all of

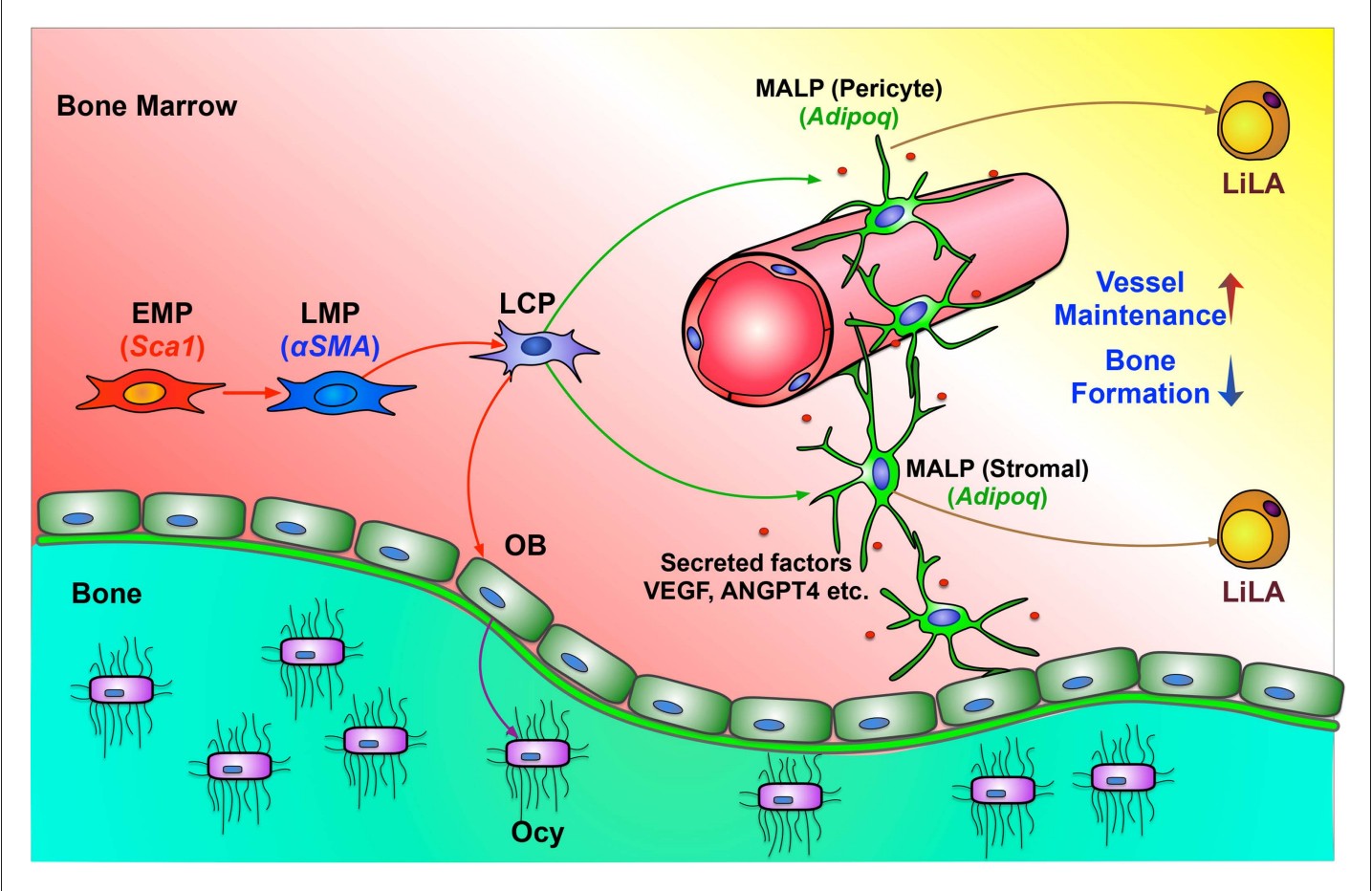

**Figure 8.** A schematic diagram depicts a model of bone marrow mesenchymal lineage cells and the role of MALPs. OB: osteoblast; Ocy: osteocyte; EMP: early mesenchymal progenitor; LMP: late mesenchymal progenitor; LCP: lineage committed progenitors; MALPs: marrow adipogenic lineage precursors; LiLA: lipid-laden adipocytes.

The online version of this article includes the following figure supplement(s) for figure 8:

**Figure supplement 1.** A comparison of large scale scRNA-seq data of bone marrow mesenchymal lineage cells between our group (Zhong et al.) and Tikhonova et al. that was recently published (*Tikhonova et al., 2019*).

**Figure supplement 2.** The comparison of large scale scRNA-seq data of bone marrow mesenchymal lineage cells between our group (Zhong et al.) and Baryawno et al. that was recently published (*Baryawno et al., 2019*) confirms the same distribution of mesenchymal subpopulations but with different annotation of cell clusters.

them. Our work computationally addresses this long-standing uncertainty and clarifies the expression patterns of previously proposed MSC markers in mesenchymal subpopulations.

It is generally believed that during organ homeostasis, a small number of quiescent, self-renewing stem cells give rise to a large number of fast-proliferating, non-renewing progenitor cells, which then mature into various terminally differentiated cell types. This unidirectionality guarantees that different cell types in an organ have distinct identities and play specialized functions. Our proposed mesenchymal progenitors at different stages and their bi-lineage differentiation routes based on scRNA-seq analysis are consistent with the above concept. Specifically, at various ages, the EMPs cluster always has the least number of cells among all mesenchymal cell clusters and is always situated at one end of trajectory. Moreover, it shrinks in size during aging. Compared to other progenitors, they are less proliferative and they do not express lineage master regulators, such as *Runx2*, *Sp7*, *Cebpa*, or *Pparg*. Instead, they express constellation of group-defining genes, including *Ly6a*, *Cd34*, and *Thy1*, which have been shown to mark other adult stem cells, such as HSCs, keratinocyte stem cells, cancer stem cells, muscle stem cells etc (*Sidney et al., 2014*; *Wilson et al., 2007*; *Shaikh et al., 2016*; *Nakamura et al., 2006*). Along the differentiation routes of progenitors, the

expression levels of lineage specific genes, such as *Bglap2*, *Col1a1*, *Ibsp*, *Adipoq*, and *Apoe*, progressively increase. The EMP cells in our dataset are also consistent with a previous study showing that mouse bone marrow Sca1$^+$PDGFRα$^+$CD45$^-$Ter119$^-$ cells generate colonies at a high frequency and differentiate into hematopoietic niche cells, osteoblasts, and adipocytes after in vivo transplantation (*Morikawa et al., 2009*). Nevertheless, future lineage tracing experiments are required to demonstrate that cells in the EMP cluster give rise to cells in LMP and LCP clusters, and resultant terminally differentiated cells in vivo.

Using a lineage tracing approach (*Lepr-Cre* mice), (*Zhou et al., 2014*) previously showed that Lepr labels bone marrow mesenchymal progenitors in adult mice (6 months of age) but not in young mice. Our study identified Lepr is a marker for MALPs, which are non-proliferative cells expressing adipocyte genes. We also observed that *Lepr* express in EMP cluster at a low level in young and adult mice and at a high level in old mice. The increased *Lepr* expression in EMP cluster during aging might explain the previous observations that Lepr marks MSCs in adult mice but not in young mice and that Lepr labels marrow adipocytes from a young age (*Zhou et al., 2014*). In the same study, it was reported that Lepr$^+$ cells constitute 0.3% of bone marrow cells. Since bone marrow CFU-F frequency is less than 0.01% in adult mice, these data clearly indicate that Lepr labels much more cells than progenitors. Our aging sequencing data showing the great expansion of Lepr$^+$ MALP cluster in 16-month-old mice further excludes the use of Lepr as a mesenchymal progenitor marker because it conflicts with the well-known aging effect on reducing the pool of bone marrow mesenchymal progenitors. Therefore, we believe that Lepr$^+$ cells in bone marrow are a heterogeneous population whose majority is MALPs and minority is mesenchymal progenitors. This is further confirmed by another scRNA-seq study analyzing *Lepr-Cre* labeled bone marrow cells (*Tikhonova et al., 2019*) (GSE108892, Portals available from http://aifantislab.com/niche). In that study, two large clusters, *Mgp*$^{High}$ P1 and *Lpl*$^{High}$ P2, highly express adipogenesis-associated genes, and two small clusters are similar to our LCP and OB clusters (*Figure 8—figure supplement 1*). A recent scRNA-seq analysis of Ter119$^-$CD71$^-$Lin$^-$bone marrow cells from *WT* mice (*Baryawno et al., 2019*) (GSE128423, Portals available from https://portals.broadinstitute.org/single_cell/study/mouse-bone-marrow-stroma-in-homeostasis) generated a similar clustering pattern and gene expression signature as ours, demonstrating the robustness of scRNA-seq methodology of identifying bone marrow mesenchymal populations. The Lepr-MSC cluster identified in that study highly and specifically expresses many known and newly identified adipocyte markers of MALPs (*Figure 8—figure supplement 2A*). Interestingly, among five fibroblast cell clusters identified in that study, 2 of them have a similar set of gene signatures as our EMPs and the other three overlap with LMPs (*Figure 8—figure supplement 2B*).

Bone marrow adipose tissue (MAT) refers to MSC-derived adipocytes located within the bone marrow niche (*Turner et al., 2018*). Different from their counterparts in white adipose tissue (WAT) and brown adipose tissue (BAT), adipocytes in MAT are interspersed at a much lower density by hematopoietic, endothelia, and osteogenic cells. To date, MAT research has focused on LiLAs and their roles in energy metabolism (*Lecka-Czernik and Stechschulte, 2014*), energy deficit (*Ghali et al., 2016*), diabetes (*Kim and Schafer, 2016*), and cancer metastasis (*Luo et al., 2018*). Since LiLAs are rare in the bone marrow of healthy young mice, MAT is mainly studied in the context of diseases and aging. Based on our computational data, we uncovered a novel type of adipogenic lineage cell, MALPs, that expresses many adipocyte markers, but lacks significant lipid stores. Our trajectory analysis, in vitro adipogenic differentiation assay, and in vivo fate mapping data provide strong evidence that MALPs represent a stable transitional cell type along the adipogenic differentiation route after mesenchymal progenitors and before LiLAs. The relationship between MALPs and LiLAs is similar to that between osteoblasts and osteocytes as both MALPs and osteoblasts are functional precursors for terminally differentiated cells. Therefore, our study expands the concept of MAT to include MALP, an abundant cell type, and its regulatory functions in bone.

In traditional adipose depots, adipogenic differentiation is tightly coupled with lipid accumulation. Single cell analyses of murine or human white fat depots (visceral or subcutaneous) did not identify cell populations analogous to MALPs (*Merrick et al., 2019*). Pre-adipocytes from adult tissues express low levels of *Pparg* but not *Adipoq*, and are also defined by their expression of canonical mesenchymal progenitor markers, including *Sca1* and *Cd34* (*Berry and Rodeheffer, 2013*). In contrast, MALPs do not proliferate and do not express progenitor markers. Instead, they express many known and newly identified markers of terminally differentiated adipocytes. Therefore, MALPs represent a cell type distinct from traditional pre-adipocytes. Under physiology conditions, MALPs

are hundreds to a thousand times more than LiLAs, indicating that they are a stable population rarely converted into LiLAs. Whether they can directly undergo apoptosis without differentiated into LiLAs needs further investigation. It will also be interesting to identify the molecular mechanism that blocks a large number of cells at MALP stage without becoming LiLAs. We speculate that MALPs are poised to accumulate lipid and become LiLAs in response to metabolic or other cues, such as radiation injury and the availability of fatty acids. Overall, we conclude that MALPs represent a marrow-specific type of adipose cell that plays unique roles in regulating their bone environment. One limitation of our study is that we do not have a *Cre* specific for MALPs. Therefore, we could not rule out the potential influence of LiLAs in cell ablation experiments. Since MALPs are hundreds times more than LiLAs in young mice, we believe that the later one has a minor role compared to the former one.

Morphologically and functionally, MALPs are also different from traditional round shape adipocytes. Even in young mice, MALPs exist in abundant amount as marrow stromal cells or pericytes. Importantly, their cell processes form a vast 3D network structure inside bone marrow, making numerous contacts among themselves and with the rest of bone marrow components. Most likely through secreting factors into their marrow environment, they play pivotal roles in maintaining marrow vasculature and suppressing osteogenic differentiation of mesenchymal progenitors. Although we did not detect any changes in hematopoietic components shortly after ablation of MALPs, we still believe that they have the ability of controlling hematopoiesis because of several reasons. First, Cxcl12, a chemokine responsible for the retention of hematopoietic progenitors (*Peled et al., 1999*), is the 2nd most expressed genes in MALPs. Previous studies have shown that Cxcl12-abundant reticular (CAR) cells act as a niche for HSCs (*Sugiyama and Nagasawa, 2012*). Indeed, due to their high Cxcl12 expression and reticular morphology, MALPs likely constitute a significant portion of CAR cells. Second, it has been shown that Scf from *Adipoq-CreER* or *Lepr-Cre* labeled bone marrow cells promotes hematopoietic regeneration after injury (*Zhou et al., 2017*). According to our datasets, Scf is highly and specifically expressed in MALPs. Hence, a prolonged cell ablation period or an injury might be required to detect the regulatory action of MALPs on hematopoiesis inside the bone marrow.

In the literature, pericytes are commonly refer to cells embedded within the basement membrane of microvessels, including capillaries, postcapillary venules, and terminal arterioles (*Armulik et al., 2011*). Our study of MALPs challenges the concept that pericytes must be tightly associated with endothelial cells. In the 3D bone marrow images of Adipoq:Td mice, while many Td+ cells display a typical pericytic morphology with extensive cell body attachment to endothelial cells, there were also Td+ cells that did not have this canonical morphology. Instead, they have a stromal cell morphology and contact endothelial cells through a small portion of their cell bodies as well as through extended processes. Moreover, many Td+ stromal cells with no cell body contact with endothelial cells extend their cell processes to wrap around capillaries walls. Therefore, these observations indicate that both pericytes and stromal cells wrap around vessels and contribute to vessel stabilization and maintenance.

The main functions of currently known adipocytes are lipid storage, thermogenesis, and regulation of nutrient homeostasis. Apparently, MALPs are unconventional possessing a completely different set of functions. Why bone marrow mesenchymal progenitors need to turn on adipocyte transcriptome in order to become a central regulator of bone marrow environment is an interesting question. Furthermore, future studying the underlying molecular mechanisms by which they regulate their environment will shed light on identifying new drugs for treating bone diseases such as osteoporosis.

# Materials and methods

## Key resources table

| Reagent type (species) or resource | Designation | Source or reference | Identifiers | Additional information |
|---|---|---|---|---|
| Genetic reagent (*M. musculus*) | *Col2-Cre* | PMID:10686612 | | |

*Continued on next page*

*Continued*

| Reagent type (species) or resource | Designation | Source or reference | Identifiers | Additional information |
|---|---|---|---|---|
| Genetic reagent (*M. musculus*) | *Dmp1-Cre* | PMID:22623172 | | |
| Genetic reagent (*M. musculus*) | *Adipoq-Cre* | PMID:21356515 | | |
| Genetic reagent (*M. musculus*) | *Acta2-CreER* | PMID:18571490 | | |
| Genetic reagent (*M. musculus*) | *Col1a1-GFP* | PMID:11771662 | | |
| Genetic reagent (*M. musculus*) | Rosa26 iDTR | Jackson Laboratory | | |
| Genetic reagent (*M. musculus*) | *Rosa26* | Jackson Laboratory | | |
| Antibody | goat anti-mouse LepR | R and D system | Cat.#: AF497 | IF(1:200) |
| Antibody | rabbit anti-mouse Laminin | Sigma | Cat.#: L9393 | IF(1:200) |
| Antibody | rat anti-mouse CD45 | Biolegend | Cat.#: 103101 | IF(1:200) |
| Antibody | rat anti-mouse Endomucin | Santa cruz | Cat.#: sc 65495 | IF(1:200) |
| Antibody | rabbit anti-Osterix | Abcam | Cat.#: ab22552 | IF(1:200) |
| Antibody | rabbit anti-Perilipin | Cell signaling | Cat.#: 9349 | IF(1:200) |
| Antibody | rat anti-mouse PDGFRβ | Biolegend | Cat.#: 136002 | IF(1:100) |
| Antibody | rabbit anti-mouse connexin 43 | Cell signaling | Cat.#: 3512 | IF(1:100) |
| Antibody | FITC rat anti-mouse Endomucin | Santa cruz biotechology | Cat.#: sc-65495 | Flow analysis (1:100) |
| Antibody | rat anti-Gr-1 APC-Cy7, | BD | Cat.#: 557661 | Flow analysis (1:100) |
| Antibody | rat anti-Mac-1 APC | eBioscience | Cat.#: 17-0112-83 | Flow analysis (1:100) |
| Antibody | rat anti-B220 FITC | eBioscience | Cat.#: 11-0452-82 | Flow analysis (1:100) |
| Antibody | hamster anti-CD3 PE-Cy7 | eBioscience | Cat.#: 25-0031-82 | Flow analysis (1:100) |
| Antibody | APC rat anti-mouse CD31 | Biolegend | Cat.#: 561814 | Flow analysis (1:100) |
| Antibody | Click-iT EdU Alexa Fluor 647 Imaging Kit | Thermo Fisher | Cat.#: D3822 | |
| Peptide, recombinant protein | Calcein | Sigma | Cat.#: C0875 | 15 mg/kg for in vivo injection |
| Chemical compound, drug | Diphtheria toxin | Sigma-Aldrich | Cat.#: D0564-1MG | 50 µg/kg for in vivo injection |
| Chemical compound, drug | ACK lysing buffer | ThermoFisher Scientific | Cat.#: A1049201 | 500 ul/ 50 million cells |
| Software, algorithm | Cellranger 3.02 | | | https://support.10xgenomics.com/single-cell-gene-expression/software/pipelines/latest/installation |
| Software, algorithm | Seurat V2, V3 | PMID:25867923 PMID:31178118 | | https://satijalab.org/seurat/ |
| Software, algorithm | Monocle V2 | PMID:24658644 | | http://cole-trapnell-lab.github.io/monocle-release/docs |

*Continued on next page*

Continued

| Reagent type (species) or resource | Designation | Source or reference | Identifiers | Additional information |
|---|---|---|---|---|
| Software, algorithm | Slingshot 1.5.1 | PMID:29914354 | | https://bioconductor.org/packages/devel/bioc/vignettes/slingshot/inst/doc/vignette.html |
| Other | BODIPY dye | ThermoFisher scientific | D3822 | IF (1:500) |

## Ethics statement

All animal work performed in this report was approved by the Institutional Animal Care and Use Committee (IACUC) at the University of Pennsylvania under Protocol 804112. University Laboratory Animal Resources (ULAR) of the University of Pennsylvania is responsible for the procurement, care, and use of all university-owned animals as approved by IACUC. Animal facilities in the University of Pennsylvania meet federal, state, and local guidelines for laboratory animal care and are accredited by the Association for the Assessment and Accreditation of Laboratory Animal Care International.

## Animal study design

*Col2-Cre Rosa26 <lsl-tdTomato>* (Col2:Td), *Dmp1-Cre Rosa26 <lsl-tdTomato>* (Dmp1:Td), *Adipoq-Cre Rosa26 <lsl-tdTomato>* (Adipoq:Td), *Adipoq-CreER Rosa26 <lsl-tdTomato>* (AdipoqER:Td), and *Acta2-CreER Rosa26 <lsl-tdTomato>* (Acta2ER:Td) mice were generated by breeding *Rosa26 <lsl-tdTomato>* (Jackson Laboratory, Bar Harbor, ME, USA) mice with *Col2-Cre* (*Ovchinnikov et al., 2000*), *Dmp1-Cre* (*Kim et al., 2012*), *Adipoq-Cre* (*Eguchi et al., 2011*), *Adipoq-CreER* (*Jeffery et al., 2014*), and *Acta2-CreER* (*Kalajzic et al., 2008*) mice respectively. *Adipoq-Cre Rosa26 <lsl-tdTomato>Col1a1-GFP* (Adipoq:Td:Col1-GFP) were generated by breeding Adipoq:Td mice with *Col1a1-GFP* mice (*Kalajzic et al., 2002*). For lineage tracing experiments, mice received Tamoxifen injections (75 mg/kg/day) at indicated age and their bones were harvested later. *Adipoq-Cre Rosa26 <lsl-tdTomato>DTR* (Adipoq:Td:DTR) mice were generated by breeding Adipoq:Td mice with *DTR* mice (Jackson Laboratory). Mice received vehicle (1xPBS) or DT injections (50 µg/kg) every other day for 2 weeks. Bones were harvested at indicated times for histology and microCT analysis.

For in vivo transplantation of mesenchymal cells, freshly FACS-sorted Td$^+$ cells (5 × 10$^4$/transplant) were mixed with Gelfoam and placed under the kidney capsule of recipient 2-month-old *C57Bl/6* mice. The transplant grafts were harvested 4 weeks post-transplantation for histology analysis. Mice received calcein injection (15 mg/kg) at 1 day prior to euthanization.

For fat transplantation, donor fat pads from *WT* siblings were cut into 100–150 mg pieces and subcutaneously implanted into 1 month old Adipoq:Td:DTR mice through small incisions in the skin with one piece per incision and six pieces per mouse as described previously (*Zou et al., 2019*; *Gavrilova et al., 2000*). Two weeks later, Adipoq:Td:DTR mice received vehicle or DT injections (50 µg/kg) every other day. At 1 week after starting DT injections, mice were starved for 6 hr followed by blood glucose measurement using a drop of tip tail blood. At 2 weeks after starting DT injections, mice were euthanized to harvest bones.

## Endosteal bone marrow Td$^+$ cell isolation and cell sorting

Endosteal bone marrow cells were harvested as described previously (*Zhu et al., 2015*). Briefly, the outer surfaces of long bones were scraped and digested to remove the periosteum. After cutting off the epiphyses and flushing out the central bone marrow, metaphyseal bone fragments were longitudinally cut into two halves and digested by proteases to collect endosteal bone marrow cells. Freshly isolated endosteal bone marrow cells were resuspended into FACS buffer containing 25 mM HEPES (Thermofisher scientific) and 2% FBS in PBS and sorted for top 1% Td$^+$ cells if a Td peak was not obvious or Td$^+$ cells if a Td peak was obvious using Influx B (BD Biosciences, San Jose, CA) or Aria B (BD Biosciences, San Jose, CA).

## Single-cell RNA sequencing of endosteal bone marrow cells

We constructed 4 batches of single cell libraries for sequencing: endosteal Td$^+$bone marrow cells from 1-month-old (n = 2), 1.5-month-old (n = 3), 3-month-old (n = 3) and 16-month-old (n = 3) male Col2:Td mice. 20,000 cells were loaded in aim of acquiring one single library of 10,000 cell for each age group by Chromium controller (V3 chemistry version, 10X Genomics Inc, San Francisco, USA), barcoded and purified as described by the manufacturer, and sequenced using a 2 × 150 pair-end configuration on an Illumina HiSeq platform at a sequencing depth of ~400 million reads. Cell ranger (Version 3.0.2, https://support.10xgenomics.com/single-cell-geneexpression/software/pipelines/latest/what-is-cell-ranger) was used to demultiplex reads, followed by extraction of cell barcode and unique molecular identifiers (UMIs). The cDNA insert was aligned to a modified reference mouse genome (mm10). Doublets or cells with poor quality (genes > 6000, genes < 200, or >5% genes mapping to mitochondrial genome) were excluded. Expression was natural log transformed and normalized for scaling the sequencing depth to a total of $1 \times 10^4$ molecules per cell, followed by regressing out the number of UMIs and percent mitochondrial genes using Seurat V2 (*Satija et al., 2015*).

For the integrated dataset, canonical component analysis (CCA) (*Butler et al., 2018*) was performed using the union of the top 2000 genes with the highest dispersion from both datasets and to determine the common sources of variation between 1- and 1.5 month datasets. A CCA dimensional reduction was subsequently generated on the basis of the first 30 canonical correlation vectors. T-distributed stochastic neighbor embedding (tSNE) plots were used to visualize the data based on the CCA alignment. Cluster specific markers for each cluster, relative to the remaining population, were conducted using the FindMarkers to identify differentially expressed genes (DEGs). Sub-clustering was performed by isolating the mesenchymal lineage clusters identified from the remaining bone marrow cells using known marker genes, followed by reanalysis as described above. This resulted in the generation of 9 clusters. Because chondrocytes are derived from the growth plate, not bone marrow, these chondrocyte clusters were excluded from further analyses. DEGs between these clusters were generated as described above. GO terms and KEGG pathway enrichment, were identified using the database for annotation, visualization and integrated discovery (DAVID) (*Huang et al., 2009*).

For individual analysis of the 3- and 16-month-old dataset, Seurat package V3 (*Stuart et al., 2019*) was used for filtering, variable gene selection, dimensionality reduction analysis and clustering standardly. Poor quality cells or doublets were first filtered as described above, followed by identifying the variable genes by default FindVariableFeatures function. Next, we performed principal component analysis (PCA) using the JackStraw function. Statistically significant PCs were selected as input for tSNE or uniform manifold approximation and projection (UMAP) plots. For sub-clustering, we repeated the same procedure of finding variable genes, dimensionality reduction, and clustering. Different resolutions for clustering have been tested to demonstrate the robustness of clusters.

To analyze 4 datasets from different age group, we applied Seurat V3 (*Stuart et al., 2019*) for performing integrated analyses to identify common cell types. 1-month-old and 1.5-month-old dataset were combined as 1M data, 3-month-old and 16-month-old dataset were input as 3M/16M data. Anchors from different dataset were defined using the FindIntegrationAnchors function, then use these anchors to integrate all datasets together with IntegrateData. This integrated data was further analyzed and subclustered for mesenchymal lineage cells as described above.

For cell cycle analysis, we used a core set of 43 S and 54 G2/M genes defined previously (*Tirosh et al., 2016*). First, the genes that are expressed in less than 5% of total cells were removed, resulting in 34 S and 45 G2/M genes for the following cell cycle analysis. Second, we define proliferative cells if the cell express one gene set significantly more than the other. Since the null distribution for the S gene set and G2/M gene set is unknown, we designed a nonparametric permutation test. To be more specific, for each cell, we resampled 34 genes from the 79 genes (34+45) as 'S genes' and the rest 45 genes as 'G2/M genes'. This resampling was repeated 5000 times and each time, we calculate the difference between the mean expression value of 'S genes' and the mean expression values of 'G2/M genes'. The original difference between the mean expression value of S genes and the mean expression value of G2M was compared with the difference between resampling results to get a significant score (p-values). To correct for multi-test, FDR corrections were applied to each

cluster. At last, FDR p-value<0.05 was used as a cutoff for distinguishing proliferative or non-proliferative cells.

To computationally delineate the developmental progression of bone marrow mesenchymal cells and order them in pseudotime, we used the algorithms implemented in the Monocle V2 package (*Trapnell et al., 2014*). We include the mesenchymal lineage cells with no chondrocytes from separated or integrated dataset of different age (1, 3 and 16 month old) mice for the analysis. We ordered cells by selecting genes with high dispersion across cells, using a parameter of 'mean_expression >= 0.05 and dispersion_empirical >= 2 * dispersion_fit', lists of genes were selected for dimensional reduction to generate the trajectory reconstruction using the nonlinear reconstruction algorithm DDRTree. Branched expression analysis modeling, or BEAM (*Trapnell et al., 2014*) was used to determine the genes that are differentially expressed between the osteogenic or adipogenic branches. A mouse transcript factors (TF) list (TFdb) (*Kanamori et al., 2004*) were used to detect 385 TFs among the differential expressed genes.

We also performed the trajectory analysis using Slingshot (*Street et al., 2018*). Briefly, UMAP was used as dimensional reduction after the PCA were calculated for individual or integrated datasets. Then Seurat objects were transformed into SingleCellExperiment objects. Slingshot trajectory analysis was conducted using the Seurat clustering information and with dimensionality reduction produced by UMAP.

## Histology

To obtain whole mount sections for immunofluorescent imaging, freshly dissected bones (femurs, tibiae, L4/L5 vertebrae) were fixed in 4% PFA for 1 day, decalcified in 10% EDTA for 4–5 days, and then immersed into 20% sucrose and 2% polyvinylpyrrolidone (PVP) at 4°C overnight. Then sample was embedded into 8% gelatin in 20% sucrose and 2% PVP embedding medium. Samples were sectioned at 50 μm in thickness. Sections were incubated with goat anti-LepR (R and D system, AF497), rabbit anti-Laminin (Sigma, L9393), rat anti-CD45 (Biolegend, 103101), rat anti-Endomucin (Santa cruz, sc-65495), rabbit anti-Osterix (Abcam, ab22552), rabbit anti-Perilipin (Cell signaling, 9349), rat anti-mouse PDGFRβ (Biolegend, 136002), rabbit anti-mouse connexin 43 (Cell signaling, 3512) at 4°C overnight followed by Alexa Fluor 488-conjugated donkey anti-goat (Abcam, ab150129), Alexa Fluor 647 anti-rat (Abcam, ab150155) or anti-rabbit (Abcam, ab150157) secondary antibodies incubation 1 hr at RT. For lipid staining, BODIPY dye (ThermoFisher scientific, D3822) was added together with secondary antibody and incubate for 1 hr at RT. For EdU staining, mice received 1.6 mg/kg EdU 1 day and 3 hr before sacrifice and the staining was carried out according to the manufacturer's instructions (ThermoFisher scientific, Click-iT EdU Alexa Fluor 647 Imaging Kit, D3822). To characterize bone marrow vessels after DT injections, all vessels in a 0.3 mm$^2$ area in the diaphyseal bone marrow were selected to measure their diameter of semi-minor axis (vessel diameter), area (vessel area), and number (vessel density).

To reconstruct 3D structure of bone marrow, fluorescence images were captured by a Zeiss LSM 710 scanning confocal microscope interfaced with the Zen 2012 software (Carl Zeiss Microimaging LLC, Thornwood, NY). Confocal image stacks were collected to a depth of ~50 μm and a step size of 1 μm at 63x magnification. Laser power and detector sensitivity were adjusted for z-correction to compensate for signal dissipation at greater imaging depths. Detector gain and offset were adjusted according to the most intense regions to ensure minimal saturation of the signal over the entire imaging area. Imaris 9.2 software (Oxford instruments, Switzerland) was used for processing z-stack and generating movies. The number of cell processes per cell was manually quantified.

## Cell culture

For CFU-F assay, unsorted endosteal bone marrow cells were plated at $1 \times 10^6$ cells/T25 flask. Sorted endosteal bone marrow cells were plated at $1 \times 10^4$, $1 \times 10^4$, and $9.8 \times 10^5$ cells/T25 flask for top 1%, 1–2%, and >2% group, respectively, based on Td intensity. Cells were cultured in growth medium (α-MEM supplemented with 15% FBS, 0.1% β-mercaptoethanol, 20 mM glutamine, 100 IU/ml penicillin, and 100 μg/ml streptomycin) for 7 days before counting CFU-F number.

Mesenchymal progenitors were obtained by culturing endosteal bone marrow cells at a high density ($3 \times 10^6$ cells/T25 flask). Once confluent, cells were cultured in adipogenic medium (DMEM with 10% FBS, 10 ng/ml triiodothyronine, 1 μM rosiglitazone, 1 μM dexamethasone, 10 μg/ml insulin, 100

IU/ml penicillin, and 100 µg/ml streptomycin) for 7 days. Brightfield and fluorescent images of mesenchymal progenitors from 1-month-old Adipoq:Td mice undergoing adipogenic differentiation were taken from day 0 to 5 by fluorescence inverted microscopy (Nikon Eclipse, TE2000-U).

For tube formation assay, we used Endothelial Progenitor Outgrowth Cells (EPOC, Biochain, Z7030001) cultured in growth medium (mouse EPOC basal medium with 10% FBS, 100 IU/ml penicillin, and 100 µg/ml streptomycin). $3.5 \times 10^5$ EPOC cells were mixed with $1.4 \times 10^5$ freshly sorted $Td^+$ cells from endosteal bone marrow of 1-month-old Adipoq:Td mice and seeded into matrigel-coated slide chamber. After 8 hr, cells were observed under fluorescence microscopy.

## Flow cytometry

Freshly isolated endosteal bone marrow cells were centrifuged to pellet cells. ACK lysing buffer (ThermoFisher Scientific, A1049201) was added to lyse red blood cells and then stained with Endomucin (FITC rat anti-mouse Endomucin, Santa cruz biotechology sc-65495) and CD31 (APC rat anti-mouse CD31, Biolegend, 561814) antibodies for 45 min on ice. After washed with flow buffer (2% FBS in PBS) twice, samples were run on LSR A and data were analyzed by FlowJo X. qRT-PCR analysis.

Sorted cells or cultured cells were collected in TRIzol Reagent (Sigma, St. Louis, MO, USA). A High-Capacity cDNAReverse Transcription Kit (Applied BioSystems, Inc, Foster City, CA, USA) was used to reverse transcribe mRNA into cDNA. Following this, quantitative realtime PCR (qRT-PCR) was performed using a Power SYBR Green PCR Master Mix Kit (Applied BioSystems, Inc). The primer sequences for the genes used in this study are listed in *Supplementary file 1*.

## Micro-computed tomography (microCT) analysis

MicroCT analysis (microCT 35, Scanco Medical AG, Brüttisellen, Switzerland) was performed at 6 µm isotropic voxel size as described previously (*Chandra et al., 2014*). At the femoral midshaft, a total of 100 slices located 4.8–5.4 mm away from the distal growth plate were acquired for trabecular bone and cortical bone analyses by visually drawing the volume of interest (VOI) separately. In the L4/L5 vertebrae, the region (total about 300 slices) 50 slices away from the top and bottom end plates was acquired for trabecular bone analysis. The trabecular bone tissue within the VOI was segmented from soft tissue using a threshold of 487.0 mgHA/cm$^3$ and a Gaussian noise filter (sigma = 1.2, support = 2.0). The cortical bone tissue was using a threshold of 661.6 mgHA/cm$^3$ and a Gaussian noise filter (sigma = 1.2, support = 2.0). Three-dimensional standard microstructural analysis was performed to determine the geometric trabecular bone volume/total volume (BV/TV) fraction, connectivity density (Conn-Dens), trabecular thickness (Tb.Th), trabecular separation (Tb.Sp), trabecular number (Tb.N), and structure model index (SMI). For analysis of cortical bone, periosteal perimeter (Ps.Pm), endosteal perimeter (Ec.Pm), porosity, cortical bone area (Ct.Ar), cortical thickness (Ct.Th), polar moment of inertia (pMOI), and tissue mineral density (TMD) were recorded.

## Hematopoietic phenotyping of bone marrow cells

Peripheral blood of mice was collected retro-orbitally. To analyze the peripheral blood of mice, red blood cells were first lysed and then stained for myeloid (rat anti-Gr-1 APC-Cy7, BD, 557661, rat anti-Mac-1 APC, eBioscience, 17-0112-83) or lymphoid lineages (rat anti-B220 FITC, eBioscience, 11-0452-82, hamster anti-CD3 PE-Cy7, eBioscience, 25-0031-82).

Bone marrow was flushed from femurs and cellularity was quantified with 3% acetic acid in methylene blue (STEMCELL). The lineage cell compartment of the bone marrow was analyzed by staining for myeloid and lymphoid lineages as in the peripheral blood. The HSPC compartment was analyzed by staining for Lineage (biotin-Ter-119, -Mac-1, -Gr-1, -CD4, -CD8α, -CD5, -CD19 and -B220 (eBioscience, 13-5921-85, 13-0051-85, 13-5931-86, 13-0112-86, 13-0452-86, 13-0041-86, 13-0081-86, 13-0193-86) followed by staining with streptavidin-PE-TexasRed (Invitrogen, SA1017), rat anti-cKit APC-Cy7 (eBioscience, 47-1171-82), rat anti-Sca1 PerCP-Cy5.5 (eBioscience, 45-5981-82), hamster anti-CD48 APC (eBioscience, 17-0481-82) and rat anti-CD150 PE-Cy7 (Biolegend, 115914). All flow cytometry analysis was performed on a BD LSR Fortessa flow cytometer and was analyzed on FlowJo v10.5.3 for MAC.

## Statistics

Data are expressed as means ± standard error (SEM) and analyzed by t-tests or two-way ANOVA with a bonferroni's post-test for multiple comparisons using Prism software (GraphPad Software, San Diego, CA). For cell culture experiments, observations were repeated independently at least three times with a similar conclusion, and only data from a representative experiment are presented. Values of $p < 0.05$ were considered significant.

## Acknowledgements

We thank Dr. Henry Kronenberg from Harvard University for critics and guidance on this study. We also thank Dr. Ivo Kalajzic from University of Connecticut Health Center for providing *Acta2-CreER* mice. This study was supported by NIH grants NIH/NIAMS R01AR066098, R01DK095803, R21AR074570 (to LQ), P30AR069619 (to Penn Center for Musculoskeletal Disorders), American Heart Association 17GRNT33650029 (to YG), R01HL095675, R01HL133828, DOD (to WT), NRSA F31HL139091 (to NH), NIH/NIDCR R00DE025915, R03DE028026 (to CC), R01DK120982 (to PS), and R00AR067283 (to ND).

## Additional information

### Funding

| Funder | Grant reference number | Author |
| --- | --- | --- |
| National Institute of Arthritis and Musculoskeletal and Skin Diseases | R01AR066098 | Ling Qin |
| National Institute of Diabetes and Digestive and Kidney Diseases | R01DK095803 | Ling Qin |
| Penn Center for Musculoskeletal Disorders | P30AR069619 | Ling Qin |
| National Institutes of Health | R21AR074570 | Ling Qin |
| American Heart Association | 17GRNT33650029 | Yanqing Gong |
| National Institutes of Health | R01HL095675 | Wei Tong |
| National Institutes of Health | R01HL133828 | Wei Tong |
| Nihon University | F31HL139091 | Nicholas Holdreith |
| National Institute of Dental and Craniofacial Research | R00DE025915 | Chider Chen |
| National Institutes of Health | R03DE028026 | Chider Chen |
| National Institutes of Health | R00AR067283 | Nathanial Dyment |
| National Institute of Diabetes and Digestive and Kidney Diseases | R01DK120982 | Patrick Seale |

The funders had no role in study design, data collection and interpretation, or the decision to submit the work for publication.

### Author contributions

Leilei Zhong, Designed the study, performed animal experiments, cell culture experiments, FACS, histology and imaging analysis, data interpretation, reviewed and revised the manuscript; Lutian Yao, Designed the study, performed animal experiments, cell culture and qRT-PCR experiments, data interpretation, performed computational analyses with assistance from ZM, RT, and ML, reviewed and revised the manuscript; Robert J Tower, Designed the study, performed animal experiments, performed computational analyses with assistance from JP, RS, and ML; Yulong Wei, Performed animal experiments; Zhen Miao, Jihwan Park, Assisted with computational analyses; Rojesh

Shrestha, Helped with library construction; Luqiang Wang, Performed cell culture and qRT-PCR experiments; Wei Yu, Performed cell culture experiments; Nicholas Holdreith, Performed flow analysis on hematopoietic cells; Xiaobin Huang, Measured blood glucose level; Yejia Zhang, Yanqing Gong, Jaimo Ahn, Fanxin Long, Provided consultation, reviewed and revised the manuscript; Wei Tong, Provided technical, material support and consultation; Katalin Susztak, Provided single cell sequence technical support; Nathanial Dyment, Provided material support; Mingyao Li, Provided technical and consultation on single cell sequence analysis; Chider Chen, Performed kidney capsule transplantation experiment; Patrick Seale, Provided technical support and consultation, reviewed and revised the manuscript; Ling Qin, Administrated the entire project, acquired resources, designed the study, performed experimental data and computational analyses, wrote the manuscript, approved the final version

## Author ORCIDs
Leilei Zhong https://orcid.org/0000-0003-1153-4115
Lutian Yao https://orcid.org/0000-0002-0652-2075
Zhen Miao https://orcid.org/0000-0002-3255-9517
Jihwan Park https://orcid.org/0000-0002-5728-912X
Wei Yu https://orcid.org/0000-0001-6705-8264
Yejia Zhang https://orcid.org/0000-0002-7484-8800
Nathanial Dyment https://orcid.org/0000-0001-8708-112X
Chider Chen https://orcid.org/0000-0003-2899-1208
Ling Qin https://orcid.org/0000-0002-2582-0078

## Ethics
Animal experimentation: All animal work performed in this report was approved by the Institutional Animal Care and Use Committee (IACUC) at the University of Pennsylvania under Protocol 804112. University Laboratory Animal Resources (ULAR) of the University of Pennsylvania is responsible for the procurement, care, and use of all university-owned animals as approved by IACUC. Animal facilities in the University of Pennsylvania meet federal, state, and local guidelines for laboratory animal care and are accredited by the Association for the Assessment and Accreditation of Laboratory Animal Care International.

## Decision letter and Author response
Decision letter https://doi.org/10.7554/eLife.54695.sa1
Author response https://doi.org/10.7554/eLife.54695.sa2

# Additional files
## Supplementary files
- Source code 1. R script for reading the matrix file of the dataset.
- Supplementary file 1. Mouse real-time PCR primer sequences used in this study.
- Transparent reporting form

## Data availability
Sequencing data have been deposited in GEO under accession code GSE145477 and the Broad Institute Cell Portal under study number SCP1017.

The following datasets were generated:

| Author(s) | Year | Dataset title | Dataset URL | Database and Identifier |
|---|---|---|---|---|
| Zhong L, Yao L, Tower RJ, Wei Y, Miao Z, Park J, Shrestha R, Wang L, | 2020 | Single cell transcriptomics analysis of bone marrow mesenchymal lineage cells | https://www.ncbi.nlm.nih.gov/geo/query/acc.cgi?acc=GSE145477 | NCBI Gene Expression Omnibus, GSE145477 |

| Author(s) | Year | Dataset title | Dataset URL | Database and Identifier |
|---|---|---|---|---|
| Yu W, Holdreith N, Zhang Y, Tong W, Gong Y, Ahn J, Susztak K, Dyment N, Li M, Long F, Chen C, Seale P, Qin L | | | | |
| Zhong L, Yao L, Tower RJ, Wei Y, Miao Z, Park J, Shrestha R, Wang L, Yu W, Holdreith N, Zhang Y, Tong W, Gong Y, Ahn J, Susztak K, Dyment N, Li M, Long F, Seale P, Qin L | 2020 | Single cell transcriptomics identifies a unique adipose lineage cell population that regulates the bone marrow environment. | https://singlecell.broad-institute.org/single_cell/study/SCP1017/single-cell-transcriptomics-identifies-a-unique-adipose-lineage-cell-population-that-regulates-the-bone-marrow-environment | Broad Institute Single Cell Portal, SCP1017 |

The following previously published datasets were used:

| Author(s) | Year | Dataset title | Dataset URL | Database and Identifier |
|---|---|---|---|---|
| Tikhonova AN, Dolgalev I, Hu H, Sivaraj KK, Hoxha E, Cuesta-Dominguez A, Pinho S, Akhmetzyanova I, Gao J, Witkowski M, Guillamot M, Gutkin MC, Zhang Y, Marier C, Diefenbach C, Kousteni S, Heguy A, Zhong H, Fooksman DR, Butler JM, Economides A, Frenette PS, Adams RH, Satija R, Tsirigos A, Aifantis I | 2019 | Bone marrow niche | https://www.ncbi.nlm.nih.gov/geo/query/acc.cgi?acc=GSE108892 | NCBI Gene Expression Omnibus, GSE108892 |
| Regev A, Scadden D | 2019 | A cellular taxonomy of the bone marrow stroma in homeostasis and leukemia demonstrates cancer-crosstalk with stroma to impair normal tissue function | https://www.ncbi.nlm.nih.gov/geo/query/acc.cgi?acc=GSE128423 | NCBI Gene Expression Omnibus, GSE128423 |

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
