## [Decision Letter]

**Acceptance summary:**

This paper elegantly uses single-cell RNA sequencing to refine our understanding of the mesenchymal components of the bone marrow. A major population of adipocyte lineage cells is found that functions to stabilize blood vessels and suppress bone formation. The findings also suggest that early multipotent progenitors may represent a much smaller fraction of the marrow than previously appreciated.

**Decision letter after peer review:**

Thank you for submitting your article "Single cell transcriptomics identifies a unique adipose cell population that regulates bone marrow environment" for consideration by *eLife*. Your article has been reviewed by three peer reviewers, including J Gage Crump as the guest Reviewing Editor and Reviewer #1, and the evaluation has been overseen by Clifford Rosen as the Senior Editor.

The reviewers have discussed the reviews with one another and the Reviewing Editor has drafted this decision to help you prepare a revised submission.

This is an interesting study that uses single-cell RNA sequencing to dissect heterogeneity of the bone marrow mesenchyme from young to adult stages in mouse. The authors report a new pre-adipocyte-like bone marrow cell population named MERA, which they define as a downstream population of MSCs by single cell RNA-seq analysis. In the first part of the paper, they computationally redefine MSCs as Sca1^+^ cells based on large-scale stromal scRNA-seq datasets. In the second part, they report that ablation of MERAs by *Adipoq-Cre* causes disruption in marrow vasculature and an increase in bone formation. Together this study is significant in clarifying the cell types in the marrow that give rise to adipocytes and osteocytes, as well as revealing a type of stromal cell with adipocyte potential that restricts bone formation and ensures proper marrow structure. What sets this study apart from preceding single-cell RNA sequencing studies is that it provides functional validation of the computational discoveries.

While the reviewers recognize the potential significance of the study, there were major concerns that would need to be addressed before it is suitable for publication.

1) The authors need to deposit their sequencing data in GEO. The data could be password protected and only accessible to the reviewers. Four out of six figures contain computational data. The reviewers found it difficult to assess the bioinformatic analysis without this raw data access.

2) Use of a new acronym "MERA" appears unwarranted as these cells have many features of previously published CXCL12 abundant reticular cells (CAR cells). Along these lines, the Morrison group described Lepr^+^ perivascular stromal population back in 2012 (Ding et al.). In 2014 Zhou et al., again from Morison's group, demonstrated that Lepr^+^ cells gave rise to adipocytes and osteoblasts. In 2019 both Baryawno et al. (Figure 2C) and Tikhonova et al. (Sub Figure 4A) reported that a Lepr-high mesenchymal population expresses high levels of Adipoq, Cxcl12, Lpl, etc. Validation of previously published data using Adipoq-Cre-TdT system does not substantiate a discovery of a novel population. Furthermore, Lepr-high cells are pericytes, which by definition are cells that wrap around the endothelial cells, and maintain homeostasis of blood vessels. The authors should better relate their findings to these previous studies, and clarify if "MERA" cells are equivalent to CAR cells or a specific CAR cell subtype. The term "MERA" should be dropped unless they clearly demonstrate why this is a previously unreported population.

3) The authors used Monocle and Slingshot to infer the ancestor of adipocytes and osteoblasts. They identified markers Sca1, Cd34, and Thy1 specific to that population. As correctly state those tools are used to infer differentiation trajectory and without convincing functional biological data MSC identification is an overstatement and should be greatly toned down. Can the authors identify Sca1+CD34+Thy1+ population by imaging? What is the location of that MSC population relative to the bone marrow? The authors should also reanalyze their datasets with UMAP-based dimensionality reduction to redefine the position of this 'MSC' cluster as tSNE is not considered a good way to visualize inter-cluster relationships. An updated version of Seurat and Monocle has this function readily available in the package. In addition, the authors might consider avoiding the use of the term "MSC" to describe cells in vivo. MSC is a historical term largely used to describe in vitro cultured cells.

4) The authors should more carefully validate the specificity of *Adipoq-CreER*. This line could accidentally mark 'MSCs' as defined in their scRNA-seq analysis. Further, as pointed out by the authors, this line also labels subcutaneous adipocytes, which can exert non-cell autonomous effects on bone formation. The study using the same *Adipoq-Cre*; DTA 'fatless' mice showed that circulating adiponectin and leptin released by subcutaneous adipocytes negatively regulate bone formation (Zou et al., 2019). The authors should highlight the caveats of this study better.

5) The definition of pericytes and stromal cells of this study is not accurate. The authors use the term pericytes to refer to the entire perivascular stromal cell populations. The general consensus is that pericytes specifically refer to a subset of perivascular stromal cells surrounding small arteries, arterioles. What they refer to as pericytes in this study are reticular cells surrounding sinusoidal vessels. Pericytes and perisinusoidal reticular cells have different morphologies and functions. The authors should clarify the characteristics of *Adipoq-Cre* labeled cells in the revised manuscript.

6) Rather than disputing the Aifantis and Scadden annotations, a suggestion would be to correlate their clusters in a more balanced way with these previous studies.

[Editors' note: further revisions were suggested prior to acceptance, as described below.]

Thank you for submitting your article "Single cell transcriptomics identifies a unique adipose cell population that regulates bone marrow environment" for consideration by *eLife*. Your article has been reviewed by three peer reviewers, including J Gage Crump as the guest Reviewing Editor and Reviewer #1, and the evaluation has been overseen by Clifford Rosen as the Senior Editor. The following individual involved in review of your submission has agreed to reveal their identity: Iannis Aifantis (Reviewer #2).

The reviewers have discussed the reviews with one another and the Reviewing Editor has drafted this decision to help you prepare a revised submission.

The reviewers agree that this is a beautiful manuscript that details a number of novel computational and biological findings that fit well to a number of closely related datasets. While the majority of issues have been addressed in this revision, I ask that you address the following concerns before we can formally accept your manuscript.

1) The reviewers all agree that use of the term MSC is problematic, especially given the lack of direct lineage data supporting the Sca1^+^ population behaving as stem cells in this study. The term "MSC" should be dropped and replaced by a more neutral term such as "mesenchymal progenitor". Perhaps "early mesenchymal progenitor" and "late mesenchymal progenitor" and stress that early/late refers to interpretation of scRNA-seq analysis rather than direct lineage tracing. Discussion of these populations as "stem cells" should also be removed throughout the manuscript.

2) Some additional language highlighting similarities of MALP cells to previously reported CAR cells should be provided.

3) The simple lineage diagram provided in the response letter (Author response image 1) would be good to include in the main Figure 1. I would also suggest using LiLA instead of Ad to denote mature adipocytes in Figure 1 and beyond.

4) I would show Author response image 2 in the paper – potentially as supplementary data. This is a nice image and confirms the presence of rare Sca1^+^ cells. However, not essential to include if authors feel otherwise.

5) Subsection “Single cell transcriptomic profiling of bone marrow mesenchymal lineage cells”: better to say "unlikely" rather than "very unlikely".

6) Should not say "mature adipocyte-specific *Adipoq-Cre* reporter" as data clearly suggest it labels a broader population. It would be better to introduce use of this *Cre* given the ability of Adipoq to label a broad population of stromal MALP cells. Rather than referring to previous publications saying *Adipoq-Cre* is a mature adipocyte marker, a more agnostic view should be taken as to what this *Cre* actually labels.

---

## [Author Response]

While the reviewers recognize the potential significance of the study, there were major concerns that would need to be addressed before it is suitable for publication.1) The authors need to deposit their sequencing data in GEO. The data could be password protected and only accessible to the reviewers. Four out of six figures contain computational data. The reviewers found it difficult to assess the bioinformatic analysis without this raw data access.

Sequencing data are now deposited in GEO. Please go to https://www.ncbi.nlm.nih.gov/geo/query/acc.cgi?acc=GSE145477.

2) Use of a new acronym "MERA" appears unwarranted as these cells have many features of previously published CXCL12 abundant reticular cells (CAR cells). Along these lines, the Morrison group described Lepr^+^ perivascular stromal population back in 2012 (Ding et al). In 2014 Zhou et al., again from Morison's group, demonstrated that Lepr^+^ cells gave rise to adipocytes and osteoblasts. In 2019 both Baryawno et al. (Figure 2C) and Tikhonova et al. (Sub Figure 4A) reported that a Lepr-high mesenchymal population expresses high levels of Adipoq, Cxcl12, Lpl, etc. Validation of previously published data using Adipoq-Cre-TdT system does not substantiate a discovery of a novel population. Furthermore, Lepr-high cells are pericytes, which by definition are cells that wrap around the endothelial cells, and maintain homeostasis of blood vessels. The authors should better relate their findings to these previous studies, and clarify if "MERA" cells are equivalent to CAR cells or a specific CAR cell subtype. The term "MERA" should be dropped unless they clearly demonstrate why this is a previously unreported population.

The concern raised by the reviewers is whether MERA is a previously unreported population. Based on our data and previous publications, we believe that MERAs are a major component of LepR^+^ and CAR cells, bearing resemblance to Adipo-CAR cells proposed in a recent publication (Baccin et al., 2020). Initial reports of LepR^+^ cells and CAR cells emphasized their stem cell/progenitor nature but ignored their heterogeneity (Omatsu et al., 2010, Zhou et al., 2014). Hence, many of their properties, such as perivascular residence and regulatory actions on hematopoiesis, were attributed to MSCs. Our study demonstrated that MERAs are neither MSCs nor progenitors with bi-lineage differentiation ability. Indeed, they are a novel adipose cell population not identified in the previous studies of LepR^+^/CAR cells. Recent studies using single cell RNA-seq confirmed that both LepR^+^ cells and CAR cells are heterogeneous, containing a large cell population with high adipocyte gene expression (Matsushita et al., 2020, Tikhonova et al., 2019). Functionally, we reported that MERAs regulate marrow vessels and bone formation as adipose cells but not as MSCs, which has never been reported for LepR^+^ cells and CAR cells.

During the Gordon Research Conference (Bones and Teeth) and ORS 2020 Annual Meeting last month, we consulted several bone and stem cell experts about the proper name for this novel cell population. We all agreed that these cells are located between mesenchymal progenitors and lipid-laden adipocytes (LiLAs) on the adipogenic differentiation route of bone marrow MSCs (Author response image 1). One suggestion is to name them Adipoq^+^ cells. However, we strongly disagree with the idea of using a marker to name a subpopulation that can be hierarchically defined in a lineage. In the past, researchers have used markers, such as PaS, LepR, Cxcl12, Osterix etc., to refer to the cells they are interested in. However, this type of nomenclature is misleading in many circumstances. For example, Zhou et al. 2014 reported that LepR^+^ cells are mesenchymal progenitors with CFU-F forming and bi-lineage differentiation abilities. Thus, they concluded that LepR^+^ cells represent the main source of bone formed by adult bone marrow. Accordingly, Baryawno et al. 2019 interpreted their LepR^+^ cell cluster as MSC. However, our data and Tikhonova et al. 2019 data clearly show that LepR^+^ cells are heterogeneous covering a wide range from MSCs to adipocytes. According to our sequencing data, LepR is expressed at a much higher level in the adipocyte cluster than in the MSC cluster. Similarly, Cxcl12+ cells are also heterogeneous covering the entire mesenchymal lineage cells with a highest level of expression in the adipocyte cluster. In addition, marker expression is often fluid and dynamic among subpopulations. For example, sequencing data revealed that LepR expression in MSC cluster increases from a very low level in young mice to a substantially high level in aging mice (Figure 3—figure supplement 2D), which fits well with the conclusion from Zhou et al. 2014 that LepR^+^ cells marks MSCs only during the adult stage but not the adolescent stage. Now with more knowledge of mesenchymal subpopulations, we believe that the hierarchical location is a better way to name a subpopulation than markers.

Another suggestion is to name them adipoblast because of the analogy between these cells and osteoblasts. Interestingly, osteoblasts are located at a similar hierarchical location as our cells in the differentiation route: between mesenchymal progenitors and osteogenic terminally differentiated osteocytes (Author response image 1). Also similar to our cells, osteoblasts are abundant and have important functions different than osteocytes.

**Author response image 1. respfig1:** Large scale scRNA-seq identifies subpopulations of bone marrow mesenchymal lineage cells from MSCs to mature cells.

The third suggestion is to drop the function-related terms and simply name them based on their hierarchical location: marrow adipogenic lineage precursors (MALPs). We adopted this suggestion and revised our manuscript accordingly. However, we also favor “adipoblast”. If the Editors/reviewers have a recommendation, we will be glad to modify our manuscript again. For now, we refer our cells as MALPs in this rebuttal letter and manuscript.

3) The authors used Monocle and Slingshot to infer the ancestor of adipocytes and osteoblasts. They identified markers Sca1, Cd34, and Thy1 specific to that population. As correctly state those tools are used to infer differentiation trajectory and without convincing functional biological data MSC identification is an overstatement and should be greatly toned down. Can the authors identify Sca1+CD34+Thy1+ population by imaging? What is the location of that MSC population relative to the bone marrow? The authors should also reanalyze their datasets with UMAP-based dimensionality reduction to redefine the position of this 'MSC' cluster as tSNE is not considered a good way to visualize inter-cluster relationships. An updated version of Seurat and Monocle has this function readily available in the package. In addition, the authors might consider avoiding the use of the term "MSC" to describe cells in vivo. MSC is a historical term largely used to describe in vitro cultured cells.

We thank the reviewers for these comments and agree that MSC is not the best term for cluster 1 cells. We added several sentences in the Results to explain that those cells are likely the most primitive cells of bone marrow mesenchymal lineage cells. We cannot exclude the possibility that there are more primitive cells that are not labeled by *Col2-Cre* but give rise to cluster 1 cells. But this possibility is low because Td expression starts from the mesenchymal condensation stage and Td labels all CFU-F colonies in Col2/Td mice. In the hematopoiesis field, HSC is defined that a single cell can re-constitute the entire blood system. Unfortunately, we do not have a similar assay in the bone field demonstrating that a single cell can generate an entire bone. Therefore, we are unable to biologically demonstrate the MSC nature of cluster 1 cells.

We also agree with the reviewers that MSC is a historical term for cultured cells. However, similar to HSC, MSC has been widely used in the literature to refer to in vivo stem cells. The most recent example is the LepR^+^ MSC cluster identified by scRNA-seq analysis of non-hematopoietic cells in bone marrow (Baryawno et al., 2019). For simplicity, we use MSCs, late MSCs, MBP, and LCP to name cluster 1, 2, 3, and 6, respectively, in the clustering plots. We will be glad to consider any other suggestions from Editors/reviewers about cluster names.

Based on the reviewers’ suggestion, we added UMAP plots to the manuscript. Clearly, UMAP generated a similar clustering pattern and gene expression patterns as tSNE, further validating the differentiation trajectory of MSCs and the existence of novel mesenchymal subpopulations we identified by tSNE.

According to the reviewers’ suggestion, we performed immunostaining of Sca1 and CD34 on 50 µm-thick Col2/Td bone sections. Td+Sca1+ or Td+CD34+ cells are rare with approximate one cell per section. They displayed a reticular stromal shape, which is different from the round shape of Td-Sca1+ or Td-CD34+ hematopoietic cells (Author response image 2). They had a stromal location with no direct contact with vessels, so they did not appear to be pericytes. Since this manuscript focuses on MALPs not MSCs and more experiments are required to confirm their MSC nature, we do not include this piece of data in this manuscript.

**Author response image 2. respfig2:** Detection of bone marrow MSCs in vivo. Representative images of stromal MSCs in 1-month-old Col2/Td mice. Arrows point to CD34+Td+ cells in A and Sca1+Td+ cells in B.

4) The authors should more carefully validate the specificity of Adipoq-CreER. This line could accidentally mark 'MSCs' as defined in their scRNA-seq analysis. Further, as pointed out by the authors, this line also labels subcutaneous adipocytes, which can exert non-cell autonomous effects on bone formation. The study using the same Adipoq-Cre; DTA 'fatless' mice showed that circulating adiponectin and leptin released by subcutaneous adipocytes negatively regulate bone formation (Zou et al., 2019). The authors should highlight the caveats of this study better.

The reviewers might refer to the Adipoq expression in cluster 1 (MSC) of tSNE plot and dropplot. Adipoq expression level is very low in MSC cluster with only 1 transcript/cell detected in 83 cells out of 280 MSCs in 1 month dataset. On the contrary, its expression level is high in the adipocyte cluster with an average 17 transcripts/cell detected in all 1077 adipocytes. It is important to note that the detection threshold of lineage tracing is likely different from that of sequencing. Our lineage tracing data clearly revealed that both *Adipoq-Cre* and *CreER* do not label CFU-F forming cells and do not incorporate EdU. Thus, they do not mark MSCs at 1 month of age.

We thank the reviewers for pointing out the data from fatless mice. Zou et al. 2019 used *Adipoq-Cre* DTA mice with no fat tissue since development. In our study, we used *Adipoq-Cre* DTR mice with normal fat tissues before DT injections. To understand whether bone phenotypes observed in our cell ablation model are due to the crosstalk from peripheral fat depots, we repeated our experiments with fat transplantation to restore peripheral fat tissues. Interestingly, fat transplants did correct hyperglycemia but did not block the abnormal vessel and bone phenotypes, indicating that MALPs, but not peripheral fat depots, affect bone marrow vasculature and bone formation. These data are now added to the manuscript (Figure 7—figure supplement 3).

5) The definition of pericytes and stromal cells of this study is not accurate. The authors use the term pericytes to refer to the entire perivascular stromal cell populations. The general consensus is that pericytes specifically refer to a subset of perivascular stromal cells surrounding small arteries, arterioles. What they refer to as pericytes in this study are reticular cells surrounding sinusoidal vessels. Pericytes and perisinusoidal reticular cells have different morphologies and functions. The authors should clarify the characteristics of Adipoq-Cre labeled cells in the revised manuscript.

Mural cells comprise two cell types required to stabilize vascular networks. Pericytes surround smaller caliber vessels, whereas vascular smooth muscle cells surround larger vessels. Also stated in a recent review (Armulik et al., 2011), “the currently accepted definition of a mature pericyte as a cell embedded within the vascular basement membrane (BM) came with the application of electron microscopy. As discussed below, this definition is difficult to apply in situations of active angiogenesis. Another commonly applied defining criterion is the presence in microvessels, i.e., capillaries, postcapillary venules, and terminal arterioles”. Therefore, pericytes refer to cells surrounding capillaries but not large vessels such as arteries and arterioles. In bone, the vast majority of vessels are capillaries. Emcn staining we used here detects capillaries but not arteries.

In our manuscript, pericytes are specifically referred to cells attached to capillaries via cell body. As shown in Figure 5, pericytes appear as small bulges on the vessels that express basement membrane Laminin. Td+ cells in Adipoq/Td bone marrow exist as pericytes and stromal cells. The later one refers to those cells not attaching to vessels via cell body. However, many of them extend cell processes to wrap around vessels as another way of contact.

We apologize for the confusion and we revised our manuscript to better reflect our definition of pericytes and stromal cells.

6) Rather than disputing the Aifantis and Scadden annotations, a suggestion would be to correlate their clusters in a more balanced way with these previous studies.

We thank the reviewers for this suggestion and we modify the Discussion accordingly.

[Editors' note: further revisions were suggested prior to acceptance, as described below.]

[…] The reviewers agree that this is a beautiful manuscript that details a number of novel computational and biological findings that fit well to a number of closely related datasets. While the majority of issues have been addressed in this revision, I ask that you address the following concerns before we can formally accept your manuscript.1) The reviewers all agree that use of the term MSC is problematic, especially given the lack of direct lineage data supporting the Sca1^+^ population behaving as stem cells in this study. The term "MSC" should be dropped and replaced by a more neutral term such as "mesenchymal progenitor". Perhaps "early mesenchymal progenitor" and "late mesenchymal progenitor" and stress that early/late refers to interpretation of scRNA-seq analysis rather than direct lineage tracing. Discussion of these populations as "stem cells" should also be removed throughout the manuscript.

We thank the reviewers for this suggestion. We agree that we cannot biologically demonstrate that Sca1^+^ cells are MSCs due to lack of suitable technique. Therefore, we change names for cluster 1-3 to early, intermediate, and late mesenchymal progenitor (EMP, IMP, and LMP), respectively. We only keep the term MSCs at the place where cited references use it.

2) Some additional language highlighting similarities of MALP cells to previously reported CAR cells should be provided.

We add a sentence in the Discussion stating that due to their high Cxcl12 expression and reticular morphology, MALPs likely constitute a significant portion of CAR cells.

3) The simple lineage diagram provided in the response letter (Author response image 1) would be good to include in the main Figure 1. I would also suggest using LiLA instead of Ad to denote mature adipocytes in Figure 1 and beyond.

We wish not to include Author response image 1 in Figure 1 based on the following reasons. First, Author response image 1 is a simplified version of Figure 8. Adding it to Figure 1 will cause redundancy. Second, Author response image 1 contains MALP, whose name has not been defined in Figure 1 yet. Later, functional studies validate its hierarchical location and reveal its function in bone, leading to the name “MALP”. In Figure 1, we use a general term Adipocyte for MALP due to their adipogenic marker expression. Figure 1 does not contain mature adipocyte LiLA because lipid-laden cells cannot be captured for single cell analysis.

4) I would show Author response image 2 in the paper – potentially as supplementary data. This is a nice image and confirms the presence of rare Sca1^+^ cells. However, not essential to include if authors feel otherwise.

We thank the reviewers for the appreciation of this figure. However, since this manuscript focuses on MALPs not Sca1^+^ cells and more experiments are required to confirm their MSC nature, we wish not to include this piece of data in this manuscript. We are currently exploring new technique to prove that they are true MSCs in vivo.

5) Subsection “Single cell transcriptomic profiling of bone marrow mesenchymal lineage cells”: better to say "unlikely" rather than "very unlikely".

We modified accordingly.

6) Should not say "mature adipocyte-specific Adipoq-Cre reporter" as data clearly suggest it labels a broader population. It would be better to introduce use of this Cre given the ability of Adipoq to label a broad population of stromal MALP cells. Rather than referring to previous publications saying Adipoq-Cre is a mature adipocyte marker, a more agnostic view should be taken as to what this Cre actually labels.

We deleted the word “mature adipocyte” throughout the manuscript.